# Neutralizing antibody activity against 21 SARS-CoV-2 variants in older adults vaccinated with BNT162b2

Joseph Newman [1,6], Nazia Thakur[1,2,6], Thomas P. Peacock[3,4], Dagmara Bialy[1], Ahmed M. E. Elrefaey [1], Carlijn Bogaardt [5], Daniel L. Horton[5], Sammy Ho[4], Thivya Kankeyan[4], Christine Carr[4], Katja Hoschler[4], Wendy S. Barclay [3], Gayatri Amirthalingam[4], Kevin E. Brown [4], Bryan Charleston[1] and Dalan Bailey [1✉]

SARS-CoV-2 variants may threaten the effectiveness of vaccines and antivirals to mitigate serious COVID-19 disease. This is of most concern in clinically vulnerable groups such as older adults. We analysed 72 sera samples from 37 individuals, aged 70–89 years, vaccinated with two doses of BNT162b2 (Pfizer–BioNTech) 3 weeks apart, for neutralizing antibody responses to wildtype SARS-CoV-2. Between 3 and 20 weeks after the second vaccine dose, neutralizing antibody titres fell 4.9-fold to a median titre of 21.3 (neutralization dose 80%), with 21.6% of individuals having no detectable neutralizing antibodies at the later time point. Next, we examined neutralization of 21 distinct SARS-CoV-2 variant spike proteins with these sera, and confirmed substantial antigenic escape, especially for the Omicron (B.1.1.529, BA.1/BA.2), Beta (B.1.351), Delta (B.1.617.2), Theta (P.3), C.1.2 and B.1.638 spike variants. By combining pseudotype neutralization with specific receptor-binding domain (RBD) enzyme-linked immunosorbent assays, we showed that changes to position 484 in the spike RBD were mainly responsible for SARS-CoV-2 neutralizing antibody escape. Nineteen sera from the same individuals boosted with a third dose of BNT162b2 contained higher neutralizing antibody titres, providing cross-protection against Omicron BA.1 and BA.2. Despite SARS-CoV-2 immunity waning over time in older adults, booster vaccines can elicit broad neutralizing antibodies against a large number of SARS-CoV-2 variants in this clinically vulnerable cohort.

The effects of the COVID-19 pandemic have, in many countries, been mitigated by the implementation of highly efficacious vaccines, which have reduced hospitalizations and deaths. Detailed information on vaccine responses in older cohorts (aged >65 years), data on neutralizing antibody titres over time and the role of boosters in enhancing cross-protection against new SARS-CoV-2 variants are lacking. Acquiring this information for older people is of especial significance because it is well established that vaccine responses in this clinically vulnerable priority group are less robust[1,2]. Our new data from a cohort of volunteers aged between 70 and 89 years who received two doses of the BNT162b2 (Pfizer) vaccine 3 weeks apart, indicate that they may be especially susceptible to repeat infection with SARS-CoV-2 variants that are antigenically distinct from the originally emerged strain; however, a third dose of BNT162b2 can mitigate against this risk.

Although the SARS-CoV-2 virus that caused the first global wave had little genetic, antigenic or other phenotypic diversity, subsequent waves comprised extremely diverse SARS-CoV-2 'variants', defined by genetic, antigenic and phenotypic divergence from preceding strains[3]. The most widespread or concerning of these were classified by the World Health Organization (WHO) and/or UK Health Security Agency (UKHSA) as 'variants of concern' (VOC), 'variants of interest/variants under investigation' (VOI, WHO; VUI, UKHSA) or 'variants under monitoring' (VUM); some VOCs and VUIs were subsequently given Greek letter identifiers[4]. Alpha (Pango lineage B.1.1.7), emerged in the UK around autumn 2020 and showed

higher transmissibility than previous variants[5,6]. Beta/B.1.351 was detected at a similar time in South Africa, with the VOC Gamma/P.1 identified in Brazil not long after[7,8]. In April 2021, Delta/B.1.617.2, spread rapidly across the world after its initial detection in India[9,10]. Most recently, from November 2021, the Omicron/BA.1/BA.2 variants first detected in southern Africa (specifically Botswana and South Africa) have risen to global prevalence. BA.1 and BA.2 show striking antigenic divergence from the original SARS-CoV-2 strains and have the highest transmissibility of any variant studied to date[11–13]. Aside from the major VOCs, many further variants have arisen throughout the world since 2020, including B.1.1.318, C.36.3, P.3/Theta, Mu/B.1.621, B.1.620, B.1.617.3, Lambda/C.37, A.30, AT.1, B.1.638 and C.1.2, with some being designated as VUIs/VUMs[14–16]. In this study, using sera from 37 volunteers aged between 70 and 89 years (recruited within UKHSA's 'COVID-19 vaccine responses after extended immunization schedules' (CONSENSUS) trial), we investigated the relative neutralization of more than 20 SARS-CoV-2 variants, correlating this data with enzyme-linked immunosorbent assays (ELISA) and receptor-binding domain enzyme-linked immunosorbent assays (RBD-ELISA) to mechanistically characterize varying antigenicity. These data provide a key tool for risk assessing current and future SARS-CoV-2 variants.

## Results

**Neutralizing antibody titres against SARS-CoV-2 wane 20 weeks after a second dose of BNT162b2.** Using a SARS-CoV-2 spike

[1]The Pirbright Institute, Guildford, UK. [2]Nuffield Department of Medicine, The Jenner Institute, Oxford, UK. [3]Department of Infectious Disease, Imperial College London, London, UK. [4]UK Health Security Agency (UKHSA), London, UK. [5]Department of Pathology and Infectious Diseases, School of Veterinary Medicine, University of Surrey, Guildford, UK. [6]These authors contributed equally: Joseph Newman, Nazia Thakur. ✉e-mail: dalan.bailey@pirbright.ac.uk

pseudotype-based micro-virus neutralization assay (mVNT), we determined neutralization titres (neutralization dose 80% ($ND_{80}$)) in 37 UK-based participants (median age 78 years (interquartile range (IQR) 75–80)) who had been vaccinated with two doses of BNT162b2 (Pfizer–BioNTech) 3 weeks apart (median 21 days (IQR 21–21); see Supplementary Dataset 1 for more details). Titres from samples taken at 3 weeks ($n = 37$, median 22 days (IQR 22–23)) and 20 weeks ($n = 35$, median 135 days (IQR 134–136.5)) after the second dose immunizations were initially evaluated with D614-based (SARS-CoV-2 lineage B (Pango)) pseudotypes to match the BNT162b2 immunogen (Supplementary Dataset 1). Dividing the cohort by age into individuals aged 70–79 ($n = 24$, median age 76 years (IQR 72–77.25)) and 80–89 ($n = 13$, median age 81 years (IQR 80–84)), we observed median titres ($ND_{80}$) of ≤128.1 (IQR 38.7–201.2) and ≤62.6 (IQR 40.9–104.8), respectively, at 3 weeks postvaccination. At 20 weeks postvaccination these titres had decreased to ≤24.84 (ages 70–79; IQR 14.9–55.7) and ≤16.0 (ages 80–89; IQR 10.0–21.3), a median reduction of ≤5.2-fold and ≤3.9-fold, respectively (Fig. 1a,b; exemplar relative light units data provided in Supplementary Fig. 1a,b and Supplementary Dataset 2). At 3 weeks after the second dose, 3/24 (12.5%) 70–79-year-olds and 3/13 (23.1%) 80–89-year-olds had no detectable neutralizing antibodies ($ND_{80} \leq 10$; limit of assay detection), increasing to 4/22 (18.2%) and 4/13 (30.8%), respectively, at 20 weeks. $ND_{80}$ titres were lower in the 80–89 age group, with a 2.0- and 1.6-fold median reduction in titre relative to the 70–79 age group at 3 and 20 weeks, respectively (see Supplementary Dataset 2 for statistical comparisons of age groups). The 3-week $ND_{80}$ titres were also converted to IU ml$^{-1}$, based on comparisons with the WHO's international standard for SARS-CoV-2 serological assays (NIBSC code: 20/136) (Supplementary Fig. 1c). The same sera samples were previously analysed[17] by ELISA (Roche Elecsys anti-SARS-CoV-2 S ECLIA) and there was a strong correlation between mVNT and ELISA titres in both age groups (70–79, Spearman $r = 0.84$, Fig. 1c; 80–89, Spearman $r = 0.91$, Fig. 1d).

**The SARS-CoV-2 Beta variant displays substantial escape from neutralization in sera from BNT162b2 doubly vaccination individuals.** The same 3-week post second dose sera were subsequently used to investigate neutralization of SARS-CoV-2 variants with epidemiological relevance using pseudotypes bearing the SARS-CoV-2 spike from VOCs Alpha (B.1.1.7), Beta (B.1.351) and Delta (B.1.617.2), as well as the D614G-containing lineage B.1, responsible for the first wave of the pandemic (Fig. 2a). Some (4/24; 16.7%) 70–79-year-olds and 5/13 (38.5%) 80–89-year-olds had no detectable neutralizing antibodies ($ND_{80} = \leq 10$; limit of assay detection) to Delta at this time point, whereas 16/24 (66.7%) and 11/13 (84.6%) 70–79- and 80–89-year-olds, respectively, had no identifiable response to Beta (Fig. 2b,c). When compared with B.1 and the 70–79 age group (median $ND_{80} \leq 333.3$; IQR 74.9–562.3) there was a ≤1.1-fold (median $ND_{80} \leq 300.0$; IQR 64.4–523.5) reduction in neutralization of Alpha, a ≤9.0-fold decrease with Delta (median $ND_{80} \leq 37.0$; IQR 17.7–67.9) and a ≤33.3-fold decrease with Beta (median $ND_{80} \leq 10.0$; IQR 10.0–14.8) (Fig. 2b and Supplementary Dataset 3). In the older age group (80–89 years) the median titres were B.1 ≤139.0 (IQR 62.1–306.4), Alpha ≤136.6 (IQR 69.1–305.5), Delta ≤12.4 (IQR 11.2–42.1) and Beta ≤10 (IQR 10.0–10.0) (Fig. 2c and Supplementary Dataset 3). For Delta and Beta this equated to a decrease in neutralizing titre of 11.2- and 13.9-fold, respectively. Neutralizing titres to Delta at 20 weeks postvaccination were slightly lower, consistent with the waning antibody response evidenced in Fig. 1, although the median value remained relatively unchanged for Beta (Extended Data Fig. 1a–d and Supplementary Dataset 3). Again, the average $ND_{80}$ titres were generally lower in the 80–89 age group compared with the 70–79 age group. VNT assays with replication-competent SARS-CoV-2 WT/D614 and the Beta VOC

also identified a reduction in neutralization for Beta, with the calculated titres correlating well with the mVNT pseudotype data (Extended Data Fig. 2). Comparing the S ELISA (Roche Elecsys anti-SARS-CoV-2 S ECLIA) with VOC $ND_{80}$ titres, we identified a strong correlation with B.1 (Spearman $r = 0.86$), Alpha ($r = 0.83$) and Delta ($r = 0.86$) neutralization titres, but a poor correlation with Beta ($r = 0.52$) (Fig. 2d). To investigate the mechanisms underpinning the decrease in neutralization seen for specific VOCs we next performed a targeted ELISA with the same sera and recombinant RBDs reflecting B.1, Alpha, Delta and Beta spike sequences. In both the 70–79 and 80–89 age groups there was no notable difference in ELISA titres between B.1 and Alpha or Delta RBDs (Fig. 2e,f); however, there was a marked reduction in binding to the Beta RBD (70–79 age group, 2.9-fold compared with B.1; 80–89 age group, 2.2-fold), partially correlating with the mVNT results. The S Roche ELISA and RBD-specific ELISA (B.1) data showed a strong correlation (Spearman $r = 0.93$), indicative of good agreement between the two assays (Fig. 2g). However, the correlation between RBD-ELISA and $ND_{80}$ titres was again poor for the Beta VOC (Spearman $r = 0.49$), albeit relatively consistent for the B.1 ($r = 0.83$), Alpha ($r = 0.80$) and Delta ($r = 0.86$) RBD-ELISA assays (Fig. 2h).

**Geographically and temporally distinct SARS-CoV-2 variants display substantial escape from BNT162b2 vaccinee sera, correlating with amino acid substitutions at spike positions 417 and 484.** Using a smaller pool of sera from the same cohort (3 weeks after the second dose; $n = 16$ total; 70–79, $n = 11$; 80–89, $n = 5$), selected by ranking the neutralization $ND_{80}$ ratio of the lineage B virus to Beta across the whole cohort and picking evenly ranked serum samples, we widened our analysis to other SARS-CoV-2 variants, including other VUIs and VUMs (Fig. 3a). This time point was chosen because the titres were higher at 3 weeks than at 20 weeks after the second dose. Several variants showed a significant reduction in neutralization when compared with B.1; namely, B.1.1.318 (2.9-fold), A.30 (2.0-fold), B.1.617.3 (1.6-fold), B.1.621/Mu (3.0-fold), P.3/Theta (7.2-fold), C.1.2 (10.8-fold), Mu + K417N (3.2-fold), B.1.638 (8.5-fold), AY.4.2 (3.4-fold) and Delta + A222V (2.8-fold) (Fig. 3b and Supplementary Dataset 4). Of note, experiments were always performed with a B.1 pseudotype control as a comparator and we showed a good concordance between calculated B.1 $ND_{80}$ titres from individual experiments, highlighting the robust repeatability of our assay and the capacity to compare mVNT titres across datasets (Supplementary Fig. 2).

To provide better spatial representation of the antigenic relationships between the different variants, we also performed antigenic cartographic analysis on the collated $ND_{80}$ titres from a subset of the sera (3 weeks after the second dose, $n = 14$) tested against all available VOCs, VUIs and VUMs in mVNTs. The antigenic map of $ND_{80}$ titres (Fig. 3c) again highlights that the largest antigenic distance is between B.1 and Beta (5.3 antigenic units (AU)). Other variants located at an intermediate distance from B.1 include C.1.2 (4.0 AU), P.3/Theta (3.5 AU), B.1.621/Mu (3.3 AU), B.1.638 (3.3 AU) and Delta (3.1 AU). The sera are all located in one part of the map, and not far from WT/D614 and B.1 viruses, as expected for sera from recipients of WT/D614 spike-based vaccines. This clustering of sera means that interpretation of the distances between the most divergent SARS-CoV-2 variants on the antigenic map (for example, Theta to Beta) is less reliable than their distances to B.1, demonstrated by the confidence coordination areas for their positions (Extended Data Fig. 3a). Lastly, to identify the amino acid changes within the RBD responsible for the striking decrease in neutralization in this cohort we re-examined the B.1, Alpha, Delta and Beta RBD-ELISA assays, extending the analysis to include RBDs with single amino acid changes at positions K417, L452, T478 or E484. The only changes that led to a notable decrease in binding were K417N (1.2-fold), E484K (1.6-fold) and E484D (1.4-fold), highlighting the

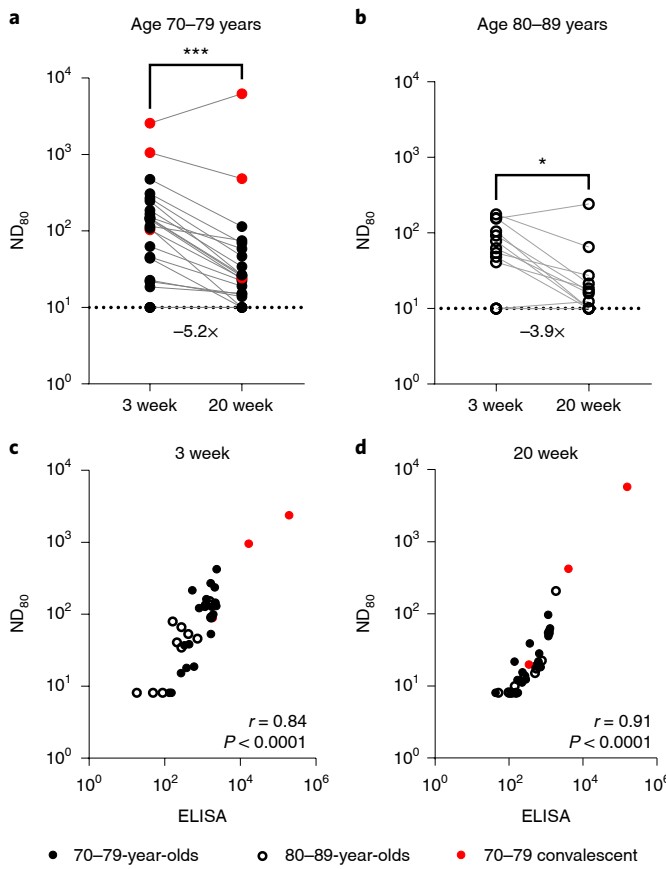

**Fig. 1 | Neutralizing antibody responses generated following BNT162b2 vaccination. a,b,** Neutralization titres calculated using pseudotypes bearing the SARS-CoV-2 D614 (lineage B) spike and sera from a cohort of BNT162b2-vaccinated individuals ($n=37$ biologically independent samples), recruited as part of the UK CONSENSUS trial, aged 70–79 ($n=24$, solid circles) (**a**) or 80–89 ($n=13$, open circles) (**b**). Symbols in red represent SARS-CoV-2 nucleoprotein-ELISA-positive samples, indicative of previous infection. Serum was collected from the same individuals at 3 ($n=37$ total) and 20 weeks ($n=35$ total) after the second dose, with a vaccination interval of 3 weeks between the first and second doses. Titres are expressed as serum fold dilution required to achieve 80% virus neutralization, with the titre ($ND_{80}$) calculated by $xy$ interpolation from the mVNT data series (dilution, $x$ versus luciferase activity, relative light units, $y$). Statistical comparison of $ND_{80}$ titres at 3 and 20 weeks was performed using a Wilcoxon two-tailed matched-pairs signed rank test (*$P < 0.05$; ***$P < 0.001$). Fold changes in median $ND_{80}$ between 3 and 20 weeks are indicated. The lower detection limit of the assay is defined as a titre of 10 (dotted line). The upper limit of detection at 3 weeks after the second dose is defined as a titre of 2,560 and at 20 weeks after the second dose as a titre of 7,290. VNTs were repeated to account for low titres and subsequent dilution series were adjusted accordingly (Supplementary Fig. 1). **c,d,** The correlation between $ND_{80}$ and S ELISA (S Roche) titres recorded from each volunteer was examined at 3 ($n=37$) (**c**) and 20 ($n=35$) (**d**) weeks after the second dose, with statistical analysis of the matrix performed using a non-parametric Spearman correlation ($r$).

importance of these positions in escape from neutralization and altered antigenicity (Fig. 3d and Extended Data Fig. 3b,c).

**A third BNT162b2 vaccine dose is effective at broadening neutralizing antibody responses, including to Omicron variants.** In November 2021 Omicron variants (BA.1/BA.2) were first detected in southern Africa, rapidly spreading across the globe. By this time,

many of the volunteers in the CONSENSUS trial had received their third vaccine dose (BNT162b2). To understand neutralization of BA.1 and BA.2 Omicron spikes, which have distinct deletions and substitutions (Fig. 4a), we performed mVNT assays for all volunteers for whom a third dose sample was available (total, $n=19$; 70–79, $n=11$; 80–89, $n=8$). These samples were taken 4 weeks after boosting (median 28 days (IQR 28–28)).

At 3 and 20 weeks after the second dose the majority of volunteers (over 80% in all age groups) had no detectable neutralizing antibodies to BA.1, although for BA.2 the situation was improved slightly (45%–75% having no detectable neutralization) (Fig. 4b and Supplementary Dataset 5). However, by 4 weeks following the third dose, all volunteers now had detectable titres against Omicron (Fig. 4b). In the 70–79 age group, when compared with B.1 (median $ND_{80} \geq 559.5$; IQR 311.5–962.7), there was a 56.0-fold reduction (median $ND_{80} \leq 10$; IQR 10.0–10.0) in neutralization of BA.1 at 3 weeks postvaccination, but only a 6.7-fold decrease (B.1; median $ND_{80} \geq 3,010.4$; IQR 1,477.9–3,555.6 versus BA.1; median $ND_{80}$ 447.3; IQR 194.0–739.8) 4 weeks after the third dose (Fig. 4b and Supplementary Dataset 5). In the 80–89 age group, the median titres for B.1 were 1,652.8 (IQR 985.4–2,868.7) after the third dose, whereas for BA.1 this was 99.0 (IQR 62.8–307.4) (Fig. 4b and Supplementary Dataset 5), a 16.7-fold reduction. For BA.2 the reduction in median $ND_{80}$ titre compared with B.1 at 3 weeks after the second dose was 42.9-fold in the 70–79 age group (median $ND_{80} \leq 10$, IQR 10.0–19.2) and 24.2-fold in the 80–89 age group (median $ND_{80} \leq 10$, IQR 10.0–10.8), but was less marked at 4 weeks after the third dose in both age groups, with only a 7.7-fold and 5.2-fold reduction compared with B.1, respectively (70–79, median $ND_{80}$ 392.8, IQR 303.6–792.1; 80–89, median $ND_{80}$ 319.0, IQR 79.3–400.7) (Fig. 4b and Supplementary Dataset 5). The median $ND_{80}$ titres between the age groups to both BA.1 and BA.2 were again lower in the 80–89 age group. Large reductions in binding to BA.1 RBD relative to B.1 were also seen, even after the third dose (92.5-fold for the 70–79 age group and 55.4-fold for 80–89 age group), consistent with the escape in neutralization reported against Omicron (BA.1 and BA.2) (Fig. 4c). Correlating and comparing the S ELISA (S1 Roche) and RBD-ELISA values for these samples with their respective BA.1 $ND_{80}$ titres again illustrated the importance of a third dose of vaccine in developing robust responses to BA.1 (Fig. 4d,e). Detectable titres against both BA.1 and BA.2 in the post third dose sera permitted antigenic cartographic analysis of this dataset, which was expanded to other VOCs (Extended Data Fig. 4 and Supplementary Dataset 5). This clearly demonstrated the large antigenic distance of BA.1 and BA.2 from B.1 (3.2 and 3.1 AU respectively; Fig. 4f), but also the effect of a third dose on reducing the antigenic distance between Beta and Delta VOCs and ancestral strains (lineage B and B.1), relative to the 3-week post second dose analysis previously performed (Extended Data Fig. 4c,d).

## Discussion

Many older adult populations have been protected from COVID-19 by the implementation of mass vaccination. However, lower overall antibody responses and concerns of waning immunity in older vaccinees have highlighted the potential impact of antigenically distinct variants on controlling this disease. Using sera from a cohort aged between 70 and 89 years who were double-vaccinated with BNT162b2 we demonstrated substantially reduced neutralization of the Beta, Delta and Omicron VOCs, decreases that are compounded by waning immunity. Before Omicron's emergence, Noori et al. performed a systematic review of the potency of BNT162b2 vaccine-induced neutralizing antibodies against SARS-CoV-2 VOCs from 36 different publications and identified a similar trend, with Beta being the least sensitive to neutralization and Delta having intermediate sensitivity[18]. Similarly, a Delta-focused review by Bian et al., examining a broader selection of studies on BNT162b2

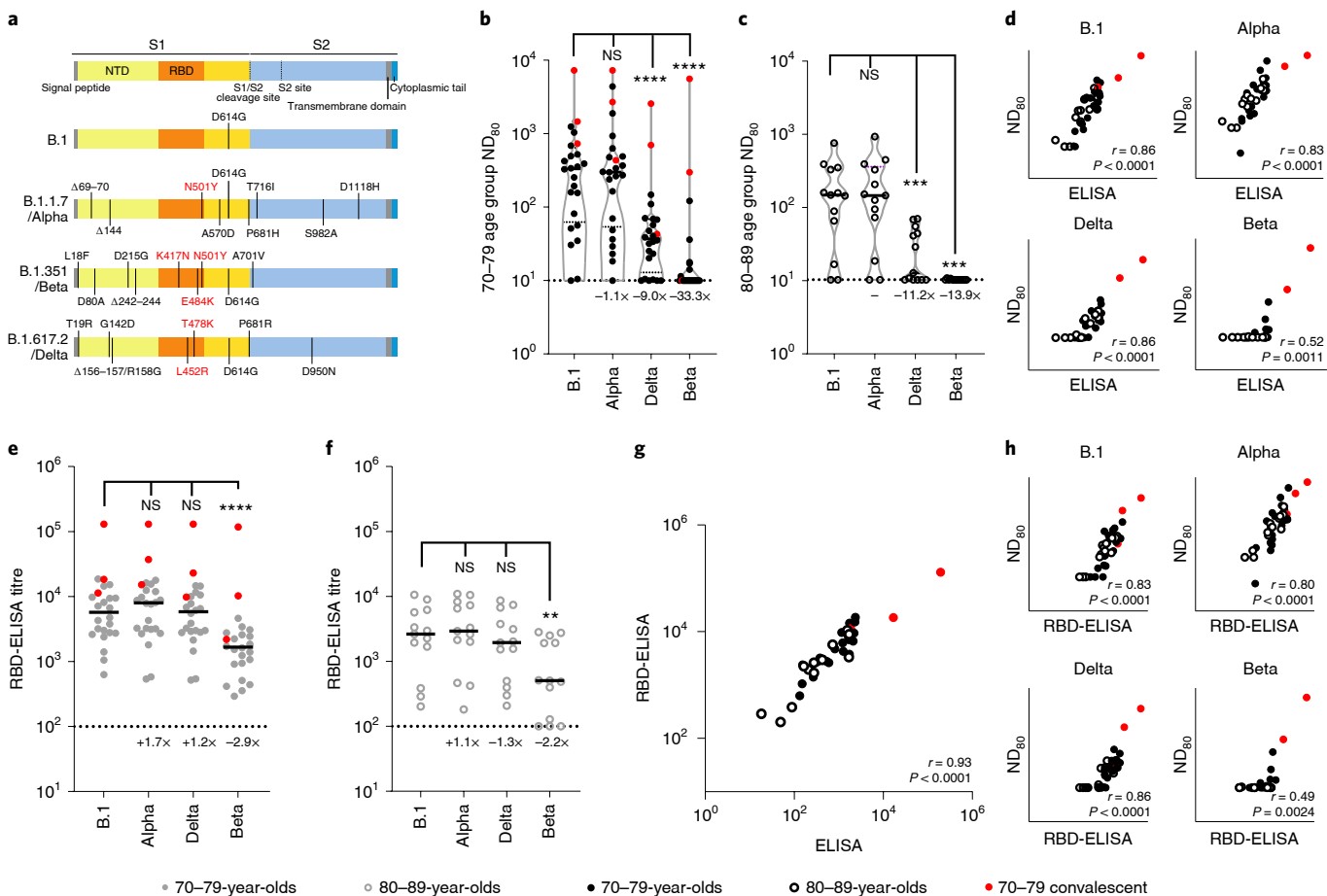

**Fig. 2 | Neutralization of SARS-CoV-2 VOC by sera collected from BNT162b2-vaccinated individuals 3 weeks after the second dose. a**, Schematic illustration of the spike mutation profiles of B.1, Alpha, Delta and Beta. NTD, N-terminal domain; RBD, receptor binding domain. **b,c**, Neutralization of pseudotypes bearing the SARS-CoV-2 B.1, Alpha, Delta or Beta spike by sera ($n = 37$ biologically independent samples) were compared in age-stratified cohorts: 70–79 years ($n = 24$, solid circles) (**b**) and 80–89 years ($n = 13$, open circles) (**c**). Statistical comparison of $ND_{80}$ titres at 3 and 20 weeks was performed using a Wilcoxon two-tailed matched-pairs signed rank test (***$P < 0.001$; ****$P < 0.0001$; NS, not significant). Fold changes in median $ND_{80}$ compared with B.1 are indicated (lower detection limit = 10 (dotted line), upper detection limit = 2,560 for Delta, 7,290 for B.1, Alpha, Beta). Medians are indicated with a solid line and upper and lower quartiles with dashed lines within the violin plots. **d–f**, The correlation between $ND_{80}$ and S ELISA (S Roche) titres for each volunteer ($n = 37$) was examined (**d**), with statistical analysis performed using a non-parametric two-tailed Spearman correlation (*r*). The same sera (70–79; $n = 24$, solid circles) (**e**) and 80–89; $n = 13$, open circles) (**f**)) was analysed with RBD-based ELISA assays, representing B.1, Alpha, Beta and Delta spikes. Horizontal lines represent the median. The equation for determining titre is described in the Methods (RBD-ELISA). Statistical comparison of RBD-ELISA titres was performed using a Friedman test with Dunn's multiple comparisons of column means (**$P < 0.005$; ****$P < 0.0001$). Fold changes in median titre compared with B.1 are indicated. The lower detection limit of the assay is defined as 100 (dotted line). The upper detection limit is defined as 129,600. **g,h**, The correlation between B.1 RBD-ELISA and S ELISA titres (S Roche) (**g**) or $ND_{80}$ titres and the respective RBD-ELISA data for each VOC (**h**) recorded from each volunteer ($n = 37$) was then examined, with statistical analysis performed using a non-parametric two-tailed Spearman correlation (*r*).

or ChAdOx1 vaccinees, as well as convalescent individuals, identified the same pattern of VOC neutralization[19]. Our data on Omicron neutralization following two doses of BNT162b2 mirrors that of Cameroni et al., Garcia-Beltran et al. and Dejnirattisai et al. who showed decreases in neutralization of >30-fold[20–23]. However, it is clear from our data, and that of Garcia-Beltran et al.[21] and Arbel et al.[24], that boosting with a third or fourth dose of BNT162b2 generates a much higher overall titre of neutralizing antibodies, enhancing cross-protection against a range of VOCs including Omicron. The number of convalescent individuals (nucleoprotein-based ELISA-positive) in this cohort was too low to be able to make meaningful comparisons with individuals who had no evidence of infection but, in general, the convalescent individuals had higher binding and neutralizing antibody titres to a broad range of variants, in agreement with other studies[25,26]. Although an ELISA-based

immune threshold that reliably correlates with a certain level of neutralizing antibodies would be advantageous for the clinical management of COVID-19, our data identified that this correlation was disrupted by antigenically distinct variants like Beta and Omicron, although this was improved after the third dose.

A smaller pool of antigenicity data is available on the wide range of other variants (VUIs/VUMs) that have emerged since the beginning of the pandemic. Our data highlighted several important antigenically divergent variants, in particular C.1.2 and B.1.638 (note, B.1.638 is from a small isolated outbreak; $n = 13$), both initially detected in South Africa[27]; B.1.621 (Mu), first detected in Colombia[28] and P.3 (Theta) isolated in the Philippines[29]. We identified a 3.0- and 10.8-fold reduction in neutralization for Mu and C.1.2, respectively (Fig. 3b), and antigenic distances of 4 AU (C1.2) and 3.3 AU (Mu) (Fig. 3c), which are concordant with observations by Tada et al. in

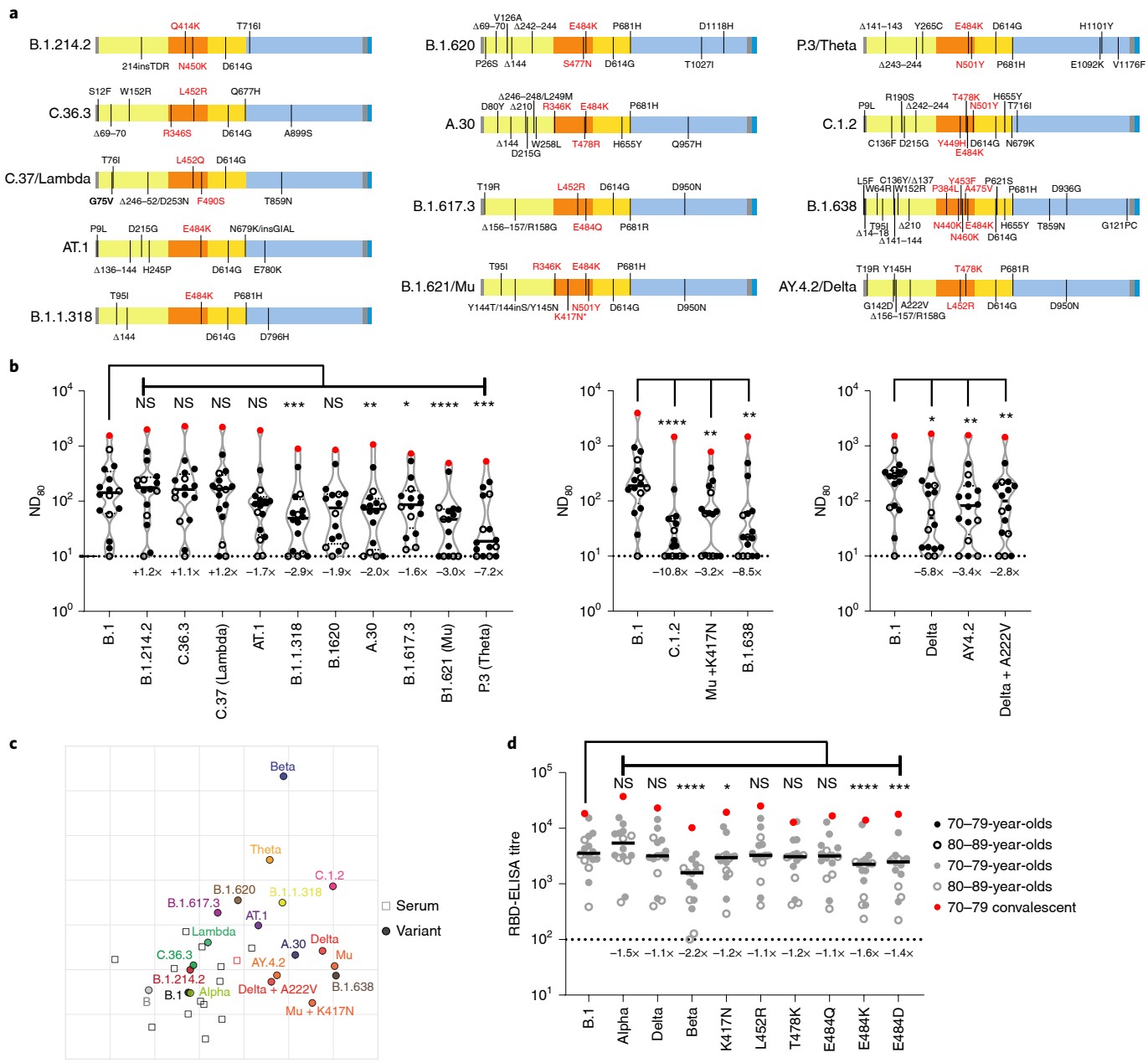

**Fig. 3 | Neutralization of a broad library of SARS-CoV-2 variants by sera collected from BNT162b2-vaccinated individuals. a**, Schematic illustration of the spike mutation profiles of 13 SARS-CoV-2 variants. The B.1.621/Mu variant has circulated with and without the K417N mutation, indicated by an asterisk. **b**, Neutralization of pseudotypes bearing these spike proteins were compared using a subsection ($n = 16$ biologically independent samples) of sera from BNT162b2-vaccinated (3 weeks after the second dose) individuals (70–79, $n = 11$, solid circles; 80–89, $n = 5$, open circles). Titration calculations and statistical analyses are as described in Fig. 1 (*$P < 0.05$; **$P < 0.005$; ***$P < 0.001$; ****$P < 0.0001$; NS, not significant). The separate graphs represent experiments performed on different days. Fold changes in median $ND_{80}$ compared with B.1 are indicated (lower limit of detection = 10 (dotted line), upper limit of detection = 7,290). **c**, Two-dimensional antigenic map of variants, based on 3-week post second dose $ND_{80}$ data. Multidimensional scaling was used to position the sera (open squares, nucleoprotein-based ELISA-positive in red) and variants (solid circles) to best fit target distances derived from the titres. Two sera were not used in mapping because of titres consistently below the lower detection limit. The spacing between grid lines represents 1 AU, equivalent to a twofold dilution in $ND_{80}$ titres. **d**, The same sera (70–79, $n = 11$, solid circles; 80–89, $n = 5$, open circles) were analysed using RBD-based ELISA assays, representing B.1, Alpha, Beta and Delta spike (data replotted from Fig. 2e,f for comparison) as well as B.1 spikes containing the individual mutations K417N, L452R, T478K, E484Q, E484K and E484D. The equation for determining titre is described in the Methods (RBD-ELISA) and the statistics are as described in Fig. 2 (*$P < 0.05$; ***$P < 0.001$; ****$P < 0.0001$). Fold changes in median titre, compared with B.1 are indicated (lower detection limit = 100 (dotted line), upper detection limit = 129,600).

BNT162b2-vaccinated individuals (Mu, 6.8-fold reduction; C.1.2, 7.3-fold)[30]. The same authors showed similar Mu and C.1.2-specific reductions in neutralization in sera from convalescent individuals as well as messenger RNA-1273 (Moderna) vaccinees[30]. There was less

agreement with the study by Uriu et al., who showed that Mu was more antigenically diverse than Beta (9.1-fold versus 7.6-fold reduction with BNT162b2 vaccine serum)[31]. Differences in fold reduction can be attributable to experimental variation (lab-to-lab, live virus

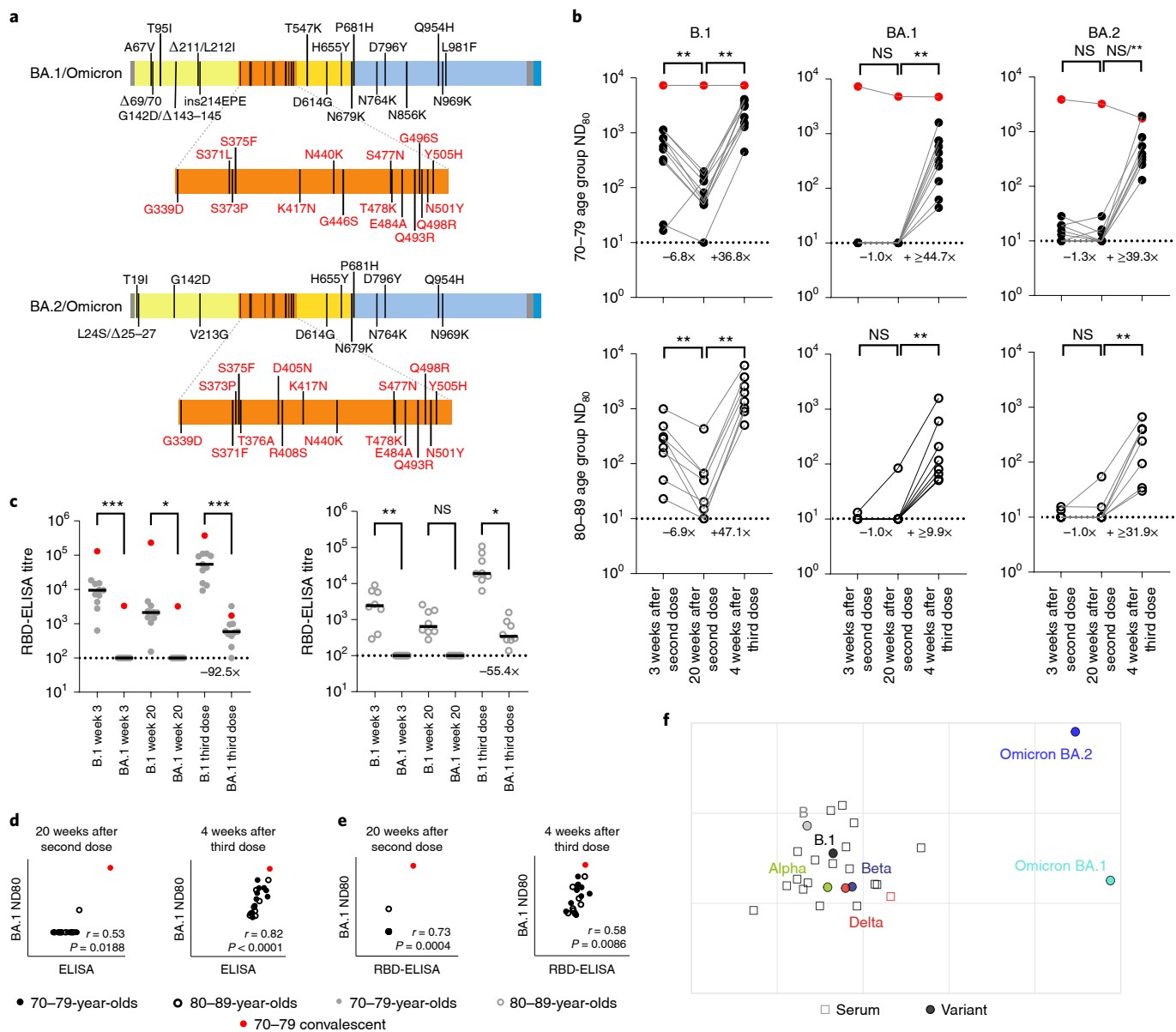

**Fig. 4 | Neutralizing antibody responses to Omicron after a third booster dose of BNT162b2. a**, Schematic illustration of the spike mutation profiles of BA.1 and BA.2 (Omicron). **b**, Neutralization titres calculated using pseudotypes bearing the SARS-CoV-2 B.1, BA.1 or BA.2 spikes and sera from a cohort of BNT162b2-vaccinated individuals at the indicated time points ($n = 19$ biologically independent samples; 70–79, $n = 11$, solid circles; 80–89, $n = 8$, open circles). Titration calculations and statistical analyses are as described in Fig. 1 (**$P < 0.005$). Note that the statistical comparison for BA.2 was significant when excluding the individual who tested positive by ELISA for SARS-CoV-2 Nucleoprotein (N) at 4 weeks after the third dose, but non-significant (NS) when included. Fold changes in median $ND_{80}$ between chronological time points are indicated (lower limit of detection = 10 (dotted line), upper limit of detection = 7,290 at 3 and 20 weeks after the second dose; lower limit of detection = 30 (dotted line), upper limit of detection = 21,870 at 4 weeks after the third dose. **c**, The same sera (70–79, $n = 11$, solid circles; 80–89, $n = 8$, open circles) were analysed using RBD-based ELISA assays (B.1 and BA.1). Solid horizontal lines represent the median. The equation for determining titre is described in the Methods (RBD-ELISA) and the statistics are as described in Fig. 2 (*$P = 0.05$; **$P < 0.005$; ****$P < 0.0001$). Fold changes in median titre compared with B.1 are indicated at 4 weeks after the third dose (lower limit of detection = 100 (dotted line), upper limit of detection = 129,600). The correlation between $ND_{80}$ and S ELISA titres (S Roche) (**d**) or RBD-ELISA (**e**) or $ND_{80}$ titres for B.1 and BA.1 was recorded from each volunteer ($n = 19$), with statistical analysis performed using a non-parametric Spearman correlation (*r*). **f**, Two-dimensional antigenic map (lowest error solution of 1,000 optimizations) of selected variants, based on 4-week post third dose $ND_{80}$ titres. Multidimensional scaling was used to position the sera (open squares, nucleoprotein-based ELISA-positive in red) and variants (solid circles) to best fit target distances derived from the titres. The spacing between grid lines represents 1 AU, equivalent to a twofold dilution in $ND_{80}$ titres.

versus pseudotype) or specific characteristics of the cohort, such as age. For example, the BNT162b2-vaccinated cohort analysed by Liu et al.[32] represents healthcare workers with a mean age of 37 (range 22–66), whereas our study targeted a cohort aged 70–89. Another interpretation is that the low neutralizing antibody titres we and

others have seen in older adult vaccinees[1,33] may contribute disproportionately to the lack of neutralization of antigenically distinct SARS-CoV-2 variants.

Our molecular understanding of SARS-CoV-2 variant spike mutations and immune escape is now relatively well established,

permitting the mechanistic contextualization of these neutralization data. The most pertinent question raised by our dataset is, 'why are Omicron and Beta so bad?', the answers to which can be used as a framework to understand other SARS-CoV-2 variants, old or new. Compared with variants such as Omicron or B.1.638, the spike mutation profile of Beta is relatively simple, with only nine changes (Fig. 3a). Focusing on the RBD, the amino acid changes K417N and E484K are well associated with antigenic escape from monoclonal antibodies and the polyclonal B-cell response[18,19,34]. Changes at position 417 affect class 1 antibody binding, while 484 modifications affect the epitope bound by class 2 antibodies, which dominate the polyclonal response to the RBD[35]. The importance of these particular RBD changes to antibody binding was confirmed by our RBD-ELISA assay (Figs. 2e,f and 3d). Combining these changes with N501Y in the Beta RBD appeared to be additive and led to a concomitant 2.2-fold reduction in antibody binding, indicating that the loss of Beta VOC neutralization we, and others[18], have observed (Fig. 2b,c) is due to a reduction in class 1 and 2 neutralizing antibody binding. A similar additive effect for E484K and N501Y was seen for Mu, Theta and C.1.2, when compared with AT.1. Although there is some evidence that N501Y modifies antigenicity[36], the presence of this mutation in Alpha does not appear to affect its neutralization[37] (Fig. 2b,c). Moreover, this substitution has most frequently been mechanistically linked to an increase in affinity for angiotensin converting enzyme 2 (ACE2)[38]. In the context of neutralization of E484-mutated variants, it may be that the increased affinity of N501Y for ACE2 has a compound effect on class 2 antibodies trying to bind spike, as these antibodies' affinities for spike have already been reduced by, for example, the E484K substitution. Interestingly, the magnitude of reduction for the complete Delta RBD (1.1-fold) or Delta-specific single amino acid RBD substitutions was lower (L452R, 1.1-fold) or the same as K417N (T478K, 1.2-fold) and non-significant in our studies (Fig. 3d). This might explain the more moderate reductions in neutralization seen with this VOC (Fig. 2b,c). Drawing conclusions on mutations that lie outside the RBD is slightly more challenging. Beta has a deletion between amino acid positions 242–244 (ref. [39]), recurrently deleted region 4 (RDR4), which corresponds to exposed loops on the surface of the spike N-terminal domain. RDR4 deletions have previously been associated with the loss of monoclonal antibody binding (4A8)[39]. In conclusion the striking antigenic escape properties of Beta and Omicron are therefore probably the result of a combination of changes to the spike protein, which act in synergy and in an additive manner to avoid antibody recognition, with the other variants we tested lacking some or all of these features.

To summarize, our data highlight the propensity of certain SARS-CoV-2 variants to partially avoid vaccine-derived immunity. The BNT162b2-vaccinated cohort we investigated showed evidence of waning immunity at 20 weeks postvaccination, which could potentially exacerbate this escape. However, high titres could be rescued with a third dose, which provided cross-protective immunity against Omicron BA.1 and BA.2. Developing a better understanding of how these titres relate to a well-defined correlate of immunity will be an important step in understanding the wider implications of this data on the management of COVID-19 and whether the risk of breakthrough infections, hospitalization and deaths is increased in older people, either by waning immunity or new variants. At a clinical level this information can be leveraged by policymakers to determine the most appropriate vaccination strategy to protect the most vulnerable groups in society.

## Methods

**Participants.** To examine antibody levels and T cell responses following the extension to the COVID-19 vaccine schedule the UKHSA (formerly Public Health England) initiated a prospective longitudinal audit of vaccinated adults (the CONSENSUS study). Healthy participants aged 70–90 years in January 2021 were recruited through North London primary care networks. Sera samples used in this study were taken at 3 and 20 weeks after two doses of Pfizer/BioNTech BNT162b2; a lipid nanoparticle-formulated, nucleoside-modified RNA vaccine encoding prefusion-stabilized SARS-CoV-2 spike[40] COVID-19 vaccine given at a 3-week interval, as well as 4 weeks after the third dose. All sera were heat-inactivated at 56 °C for 2 h before use. With regards to interpretation of the data, initially in the UK, for BNT162b2, two doses of this vaccine were administered 3 weeks apart, with many of the most clinically vulnerable (within the nine priority groups established by the UK's Joint Committee on Vaccination and Immunization) receiving their vaccines with this dosing interval. However, in the UK, this schedule was quickly changed to 'up to 12 weeks', to maximize use of limited supplies of these vaccines and to protect the largest possible number of people from developing serious disease. This remained the strategy as vaccination was extended to the priority groups further down Joint Committee on Vaccination and Immunization's list (stratification based primarily on age), before being opened up to all adults later in 2021, as well as children over 12. Third doses, as well as boosters, are now available to all adults[41].

**Cells.** HEK293T cells were used to generate lentiviral pseudoparticles bearing the SARS-CoV-2 spike. HEK293T cells stably expressing human angiotensin converting enzyme 2 under 1 μg ml⁻¹ puromycin (Gibco) selection were used for pseudotype neutralization assays[42]. Vero-E6-TMPRSS2 cells (gift from S.J.D. Neil, King's College London) under 400 μg ml⁻¹ G418 (Gibco) selection were used for live virus neutralization assays. Cells were maintained in DMEM (Sigma-Aldrich) supplemented with 10% FBS (Life Science Production), 1% 100 mM sodium pyruvate (Sigma-Aldrich), 1% 200 mM L-glutamine (Sigma-Aldrich) and 1% penicillin/streptomycin (10,000 U ml⁻¹; Life Technologies) at 37 °C in a humidified atmosphere of 5% $CO_2$.

**Plasmids.** Mutant SARS-CoV-2 expression plasmids were generated by site-directed mutagenesis, using the QuikChange Lightning Multi Site-Directed Mutagenesis Kit (Agilent) or were synthesized by Geneart (Thermo Fisher Scientific) (primer list in Supplementary Datasheet 6). All SARS-CoV-2 spike expression plasmids were based on a codon optimized SARS-CoV-2 lineage B (Pango) reference sequence (GenBank ID NC_045512.2)[43], with the additional substitutions K1255*STOP (also referred to as Δ19 mutation or cytoplasmic tail truncation). A list of the SARS-CoV-2 spike variants and their associated mutations can be found in Supplementary Datasheet 7. Some substitutions here differ from the lineage-defining sequence for the named variant (Pango); these substitutions were included because they were highly sampled in submitted sequences and predicted to be a plausible worst case antigenic escape for that lineage.

**Generating lentiviral-based pseudotypes bearing the SARS-CoV-2 spike.** Lentiviral-based SARS-CoV-2 pseudotyped viruses were generated in HEK293T cells incubated at 37 °C, 5% $CO_2$, as previously described[44]. Briefly, cells were transfected with 900 ng of SARS-CoV-2 spike (Supplementary Datasheet 7), 600 ng of p8.91 (encoding for HIV-1 gag-pol) and 600 ng of CSFLW (lentivirus backbone expressing a firefly luciferase reporter gene) with polyethylenimine (1 μg ml⁻¹) transfection reagent. Supernatants containing pseudotyped SARS-CoV-2 were harvested and pooled at 48 and 72 h post-transfection, centrifuged at 1,300g for 10 min at 4 °C to remove cellular debris and stored at −80 °C. SARS-CoV-2 pseudoparticles were titrated on HEK293T cells stably expressing human ACE2 and infectivity was assessed by measuring luciferase luminescence after the addition of Bright-Glo luciferase reagent (Promega) and read on a GloMax-Multi+ Detection System (Promega).

**mVNT using SARS-CoV-2 pseudoparticles.** Sera were diluted in serum-free media in a 96-well plate in triplicate and titrated threefold. The starting sera dilution was adjusted depending on the expected neutralization titres. A fixed titred volume of SARS-CoV-2 pseudoparticles was added at a dilution equivalent to 10⁵–10⁶ signal luciferase units in 50 μl of DMEM-10% and incubated with sera for 1 h at 37 °C and 5% $CO_2$. Target cells expressing human ACE2 were then added at a density of $2 \times 10^4$ in 100 μl and incubated at 37 °C, 5% $CO_2$ for 48 h. Firefly luciferase activity was then measured after the addition of Bright-Glo luciferase reagent on a GloMax-Multi⁺ Detection System (Promega). Pseudotyped virus neutralization titres were calculating by interpolating the point at which there was an 80% reduction in reduction in luciferase activity, relative to untreated controls ($ND_{80}$).

**RBD-ELISA.** Antibody against the RBD of the SARS-CoV-2 spike protein was measured using an in-house indirect immunoglobulin G (IgG) RBD assay[45,46]. Briefly, commercially synthesized recombinant RBD subunit spike (Arg319-Phe541(V367F); SinoBiological) with a C-terminal mouse Fc tag, was coated onto 96-well microtiter plates at 20 ng per well at 4–8 °C for a minimum of 16 h. After washing and blocking, sera were analysed at a dilution factor of 1 in 100 by serially diluting each serum sample starting at 1:100 (sixfold with the highest dilution achieved 129,600). The binding of IgG on the plate surface was detected using an anti-human IgG horseradish peroxidase antibody conjugate diluted 1:15,000 (Sigma-Aldrich, catalogue no. AP112P) and 3,3′,5,5′-tetramethylbenzidine (Europa Bioproducts). We analysed samples in

duplicate and evaluated the optical density (OD) at 450 nm. Samples were analysed in the presence of known positive controls (collected from individuals with confirmed SARS-CoV-2 infection) and a calibrator sample ('negative' added to four wells; collected before the pandemic). Titres are expressed as serum fold dilution required to achieve a T/N ratio (test OD to negative OD) of 5 (T/N = 5 serves as cut-off for positive samples) by $xy$ interpolation from the RBD data series (dilution, $x$ versus $OD_{450}$, $y$). Samples that were below the cut-off in the initial dilution (negative), were expressed as <100.

**ELISA N and S Roche.** Sera samples were tested by commercial ELISA[17]. Nucleoprotein (N) antibodies were measured as a marker of previous SARS-CoV-2 infection (Anti-SARS-CoV-2 total antibody assay; Roche Diagnostics) and spike (S) protein antibodies were measured as an indication of infection or vaccination (Elecsys Anti-SARS-CoV-2 S total antibody assay; Roche Diagnostics: positive ≥0.8 arbitrary units per ml to assess vaccine response).

**Infectious SARS-CoV-2 VNT.** Sera were serially diluted 1:2 in media containing 1% FCS and incubated with 112.5 plaque-forming units (p.f.u.) of SARS-CoV-2 (hCoV-19/England/02/2020, EPI_ISL_407073, or Beta (B.1.351), kindly provided by Public Health England) for 1 h at 37 °C, 5% $CO_2$. Neutralization mixture (75 p.f.u. in a total volume of 200 μl) was then added to 96-well plates containing an approximately 80%–90% confluent monolayer of Vero-E6-TMPRSS2 (gift from S.J.D. Neil, Kings College London) cells were incubated for 6 d at 37 °C, 5% $CO_2$ in quadruplicate per serum sample. Inoculum was then removed, and cells were fixed with formalin for 30 min before staining with 0.1% toluidine blue in PBS. $ND_{80}$ titres were calculated using a Spearman and Karber formula. Thawed virus aliquots used for VNT were back-titrated 1:10 by calculation of 50% tissue culture infectious dose on Vero-E6-TMPRSS2 cells to confirm the titre at time of use.

**Cartography.** Antigenic cartography allows high-resolution quantitative comparison and visualization of antigenic relationships. Antigenic distances between antigens on the map are measured in antigenic units, with 1 AU being the equivalent of a twofold dilution in titre. Antigenic maps were made using the antigenic cartography techniques described previously[47], implemented in the R package *Racmacs* (v.1.1.12; (after second dose $ND_{80}$ and ELISA maps) and v.1.1.18 (after third dose $ND_{80}$ map) https://acorg.github.io/Racmacs/). In brief, titres were first converted into serum–antigen target distances, and sera and antigens were then positioned on a map in a way that minimized the difference (residual sum of squares) between target distances and corresponding map distances, using multidimensional scaling. The target distance for each serum–antigen pair was calculated by subtracting $\log_2$(titre) from the highest $\log_2$(titre) for the serum against an antigen; thus, higher reciprocal titres resulted in shorter target distances. Multidimensional scaling was carried out with 1,000 random restart optimizations, to avoid local optima and increase the likelihood of finding the best fit to the measured titres. The resulting maps were ordered according to total error, and compared for self-consistency; the figures and descriptions in this article pertain to the maps with the lowest total error. Antigenic distances were measured from the lowest error antigenic map.

The antigenic map of the 3-week post second dose $ND_{80}$ titres was made based on titres from a subset of 14 sera: two sera were removed before mapping because their titres were consistently below or only marginally above the detection limit, and they did not therefore contain valuable information for cartography. $ND_{80}$ titres from different experiments were merged without normalization procedures. For each serum, a single overall $\log_2$(titre) to B.1 and B.1.617.2/Delta was calculated by taking the mean of the $\log_2$(titre) values for these antigens across experiments. Such average $\log_2$(titre) values were excluded (replaced with NA) in case the standard deviation of $\log_2$(titre) values for the antigen was equal to or exceeded 1.

To determine the optimal number of dimensions for representing the data, prediction experiments were performed: antigenic maps were made while omitting a random 10% of titres. The excluded titres were predicted according to their relative positions in the map, and the predicted titres were then compared with the actual titres (on a log scale). Antigenic maps were made in two, three, four and five dimensions, using 1,000 optimizations; for each dimension, 100 prediction tests were performed. The mean root mean square error (r.m.s.e.) associated with the prediction of 3-week post second dose $ND_{80}$ titres in a two-dimensional (2D) map was 0.94 (s.d. 0.15) for detectable titres and 1.32 (s.d. 0.68) for titres below the limit of detection; for the prediction of 4-week post third dose $ND_{80}$ titres in a 2D map this was 0.72 (s.d. 0.17) for detectable titres and 2.62 (s.d. 0.18) for titres below the limit of detection. For the prediction of RBD-ELISA titres in a 2D map this was 0.55 (s.d. 0.16) for detectable titres and 1.27 (s.d. 0.13) for titres below the limit of detection. Overall, for each dataset, the mean r.m.s.e. was similar across dimensions, and in each case the mean r.m.s.e. for prediction of detectable titres corresponded to less than a twofold dilution (1 AU). Therefore, we considered 2D maps sufficient for representing the SARS-CoV-2 antigenic data. Differences in positions of key viruses before and after the third dose were visualized using a Procrustes transformation in which maps are rotated and translated to minimize the error between maps, before overlaying one map onto the other.

**Statistical analysis.** All statistical analyses were performed using GraphPad Prism v.9 (GraphPad Software) and Excel (v.2111, Microsoft 365). Estimates of

standard deviation were computed using 'Sample' standard deviation formulas, and interquartile ranges were inclusive of all data points. For statistical comparisons of paired sets of data, a Wilcoxon matched-pairs signed rank test was used; where the data were unpaired, a Mann–Whitney test was used. To determine statistical significance for the RBD-ELISA, a Friedman test and Dunn's test for multiple comparisons were used. To measure correlation between VNT:ELISA and ELISA:ELISA, Spearman's rank correlation was used. Where most samples were at the limit of detection, statistical comparisons were not performed. A $P$-value <0.05 was deemed statistically significant. No statistical method was used to predetermine sample size. The experiments were not randomized, and the investigators were not blinded to sample groups during experiments.

**Ethics statement.** The protocol was approved by Public Health England Research Ethics Governance Group (reference NR0253; 18/01/21).Informed consent was obtained from all participants and individuals who were unable to provide informed written consent were excluded from the recruitment process.

**Reporting summary.** Further information on research design is available in the Nature Research Reporting Summary linked to this article.

## Data availability
Datasets generated and/or analysed during the current study are appended as supplemental datasheets. Only aggregated data regarding participants from the CONSENSUS trial is available due to participant confidentiality which states that individual participants will not be identified, and individual results will not be reported. All SARS-CoV-2 spike expression plasmids were based on a codon optimized SARS-CoV-2 lineage B (Pango) reference sequence (GenBank ID NC_045512.2). Source data are provided with this paper.

## Code availability
No custom code was used in the analyses provided.

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

## Acknowledgements

We would like to acknowledge the whole of the UK Health Security Agency (formerly Public Health England) CONSENSUS team, as well as the vaccinated volunteers for their help and participation in supporting this study. This work was supported by the Medical Research Council-funded G2P-UK National Virology Consortium; G2P-UK; A National Virology Consortium to address phenotypic consequences of SARS-CoV-2 genomic variation (MR/W005611/1). J.N., N.T., D. Bialy, A.M.E.E., B.C. and D. Bailey were also funded by The Pirbright Institute's Biotechnology and Biological Sciences Research Council institute strategic programme grant (BBS/E/I/COV07001, BBS/E/I/00007031, BBS/E/I/00007038, BBS/E/I/00007039 and BBS/E/I/00007034) with N.T. receiving studentship support from BB/T008784/1. C.B. and D.L.H. were supported by funding from the European Union's Horizon 2020 Research and Innovation programme under grant agreement No. 773830: One Health European Joint Programme. We would also like to acknowledge the National Institute for Communicable Diseases (NICD) and the KZN Research Innovation and Sequencing Platform (KRISP), as part of the Network for Genomic Surveillance in South Africa, for depositing the B.1.638 sequences, and the NICD, in particular D.G. Amoako and J. Everatt for providing feedback on the manuscript. The sequencing activities at the NICD were supported by: a conditional grant from the South African National Department of Health as part of the emergency COVID-19 response; a cooperative agreement between the NICD of the National Health Laboratory Service and the UK Department of Health and Social Care, managed by the Fleming Fund and performed under the auspices of the SEQAFRICA project.

## Author contributions

D. Bailey, J.N. and N.T. conceived and planned experiments. J.N., N.T., T.P.P., D. Bialy, A.M.E.E., C.B., S.H., T.K. and C.C. performed the experiments. G.A. and K.E.B. organized the CONSENSUS study and provided samples. D.L.H., K.H., W.S.B. and D. Bailey provided supervision. D. Bailey, W.S.B. and B.C. acquired funding. D. Bailey, J.N. and N.T. wrote the manuscript with input from all authors.

## Competing interests

The authors declare no competing interests.

## Additional information

**Extended data** is available for this paper at https://doi.org/10.1038/s41564-022-01163-3.

**Correspondence and requests for materials** should be addressed to Dalan Bailey.

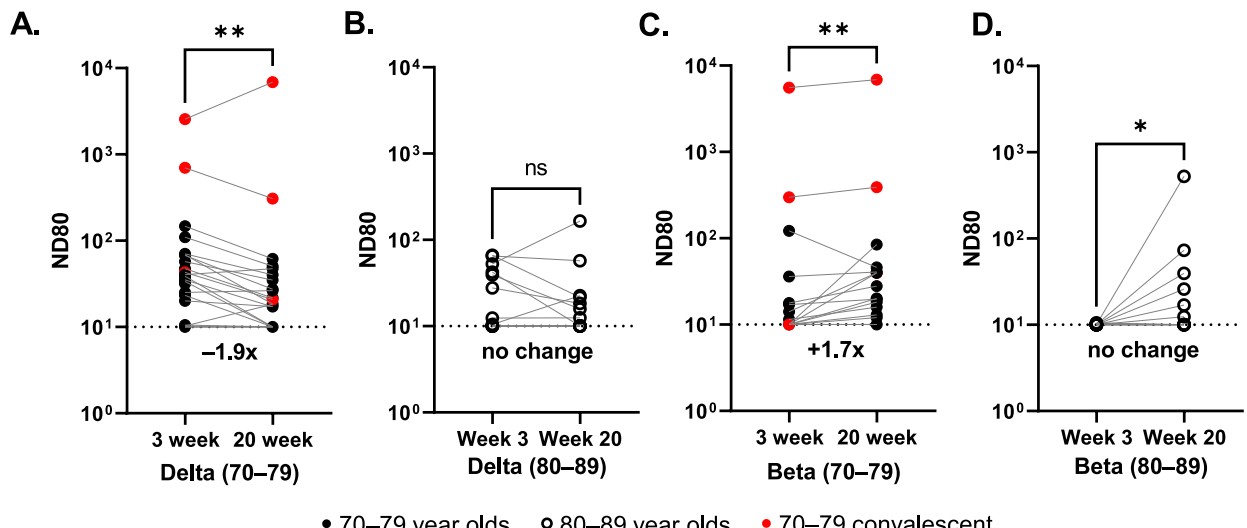

**Extended Data Fig. 1 | Comparison of neutralising antibody titres against SARS-CoV-2 VOCs at 3- and 20-weeks post 2nd dose of BNT162b2.**
Neutralisation of pseudotypes bearing the SARS-CoV-2 Delta or Beta Spike were compared in the two age-stratified cohorts, 70–79 (solid circles) (**a,c**) and 80–89 (open circles) (**b,d**). Sera for mVNTs was collected from the same individuals at 3- ($n=37$ total biologically independent samples) and 20-weeks ($n=35$ total biologically independent samples) post 2nd dose. Symbols in red represents samples taken from individuals who tested positive for SARS-CoV-2 Nucleoprotein by ELISA, indicative of previous infection. Titres are expressed as serum fold-dilution required to achieve 80% virus neutralisation, with the titre ($ND_{80}$) calculated by $xy$ interpolation from the mVNT data series (dilution, $x$ versus luciferase activity, relative light units, $y$). The detection limit of the assay is defined as a titre of 10 and is indicated with a dotted line. The upper limit of detection is defined as a titre of 2560 for Delta and 7290 for Beta. Statistical comparison of $ND_{80}$ titres at 3 and 20 weeks was performed using a Wilcoxon two-tailed matched-pairs signed rank test (*; <0.05; **, <0.01). Fold changes in median $ND_{80}$ between 3 and 20 weeks are indicated.

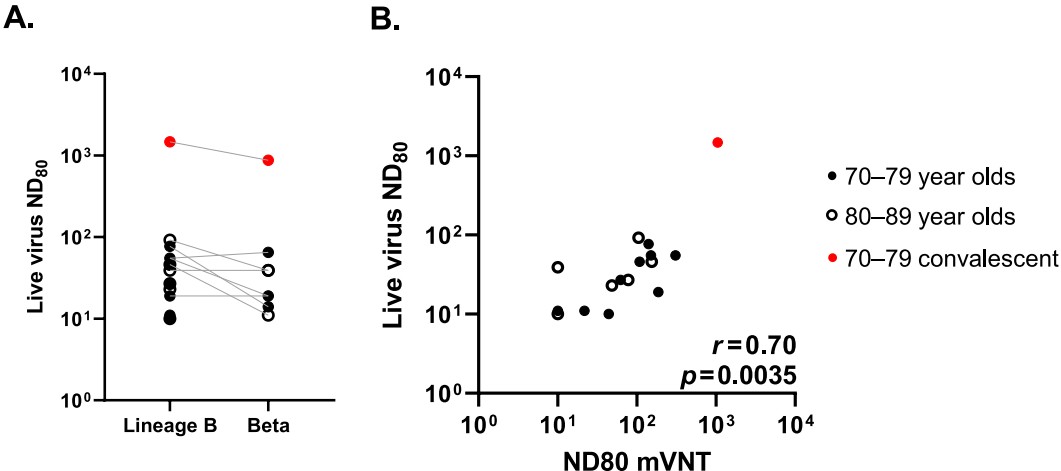

**Extended Data Fig. 2 | Comparison of neutralising antibody titres using pseudotype and live SARS-CoV-2 VOCs. a**, $ND_{80}$s were calculated using sera from a sub-section ($n=16$ biologically independent samples) of the BNT162b2-vaccinated cohort (ages 70–79, solid circles; 80–89, open circles) and live SARS-CoV-2 virus isolates (D614 [Lineage B] or Beta). **b**, The corresponding $ND_{80}$s calculated using pseudotypes (Fig. 2) were compared to these live virus $ND_{80}$s, with statistical analysis of the matrix performed using a nonparametric Spearman correlation ($r$). The limit of detection is defined as a titre of 10. Symbols in red represents samples taken from individuals who tested positive for SARS-CoV-2 Nucleoprotein by ELISA, indicative of previous infection.

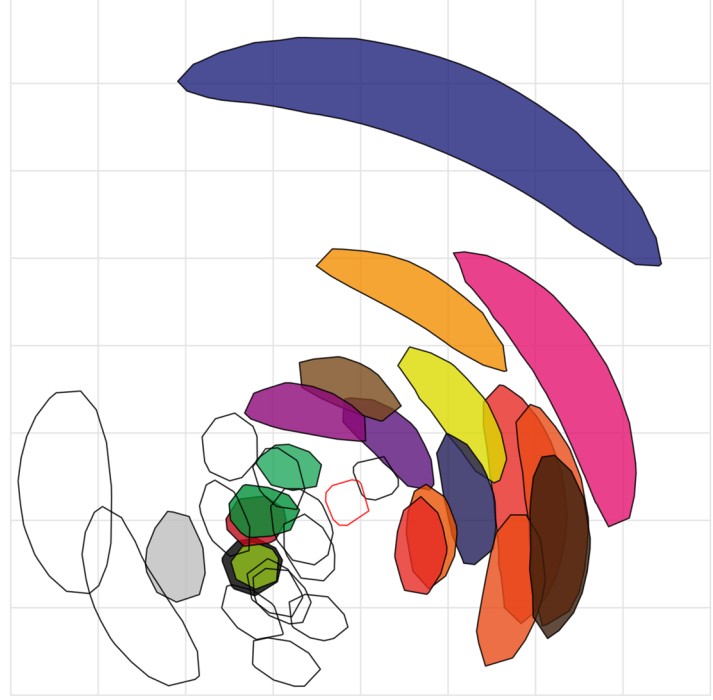

**A.**

**B.**

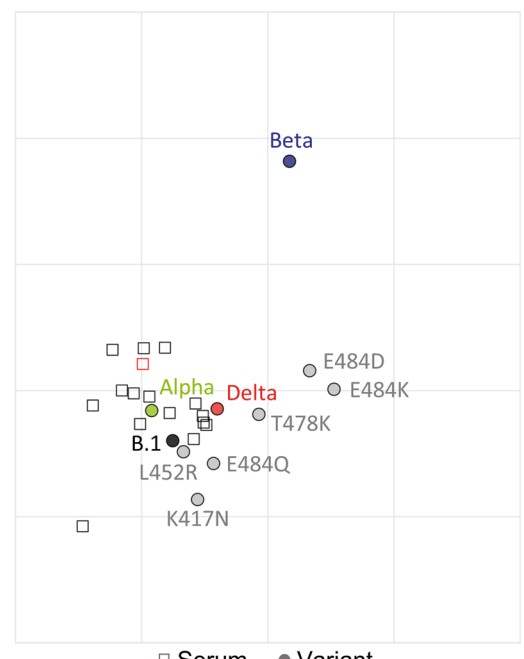

Beta

E484D

Alpha    Delta

E484K

B.1

T478K

L452R    E484Q

K417N

☐ Serum    ● Variant

**C.**

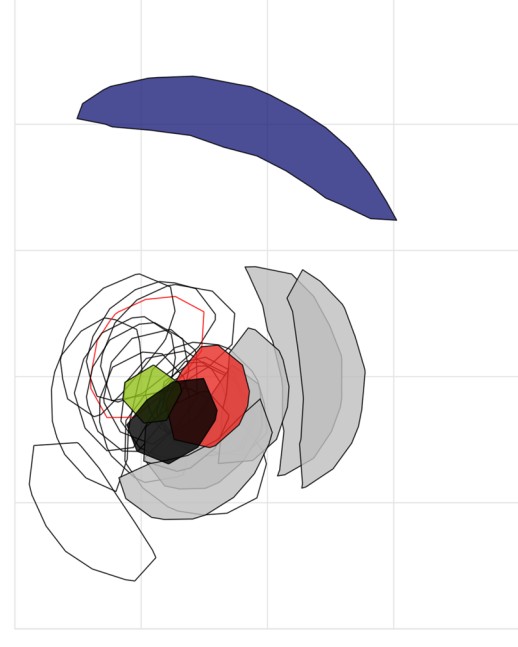

**Extended Data Fig. 3 | Antigenic cartography for variant Spike RBDs. a**, Two-dimensional antigenic map of variants (the same map as in Fig. 3c), with coordination confidence areas reflecting the uncertainty in positioning of variants and sera. Variants are represented by solid shapes, sera by open shapes (Nucleoprotein-ELISA-positive in red). Each shape encompasses the area on the map that the point could be located at, without increasing the error of the map by more than 1. The spacing between grid lines represents one antigenic unit, equivalent to a two-fold dilution in $ND_{80}$ titres. **b**, Two-dimensional antigenic map of variant Spike RBDs, based on the RBD-ELISA titres in Fig. 3d. Multidimensional scaling was used to position the sera and Spike RBDs to best fit target distances derived from the titres. The map is the lowest error solution of 1000 optimisations. Spike RBDs are represented by solid circles (variants coloured as in Fig. 3a, RBDs with individual mutations in grey), sera by open squares (Nucleoprotein-ELISA-positive in red). The spacing between grid lines represents one antigenic unit, equivalent to a two-fold dilution in RBD-ELISA titres. **c**, The same map as **b**, with coordination confidence areas reflecting the uncertainty in positioning of Spike RBDs and sera. Each shape encompasses the area on the map that the point could be located at, without increasing the error of the map by more than 1.

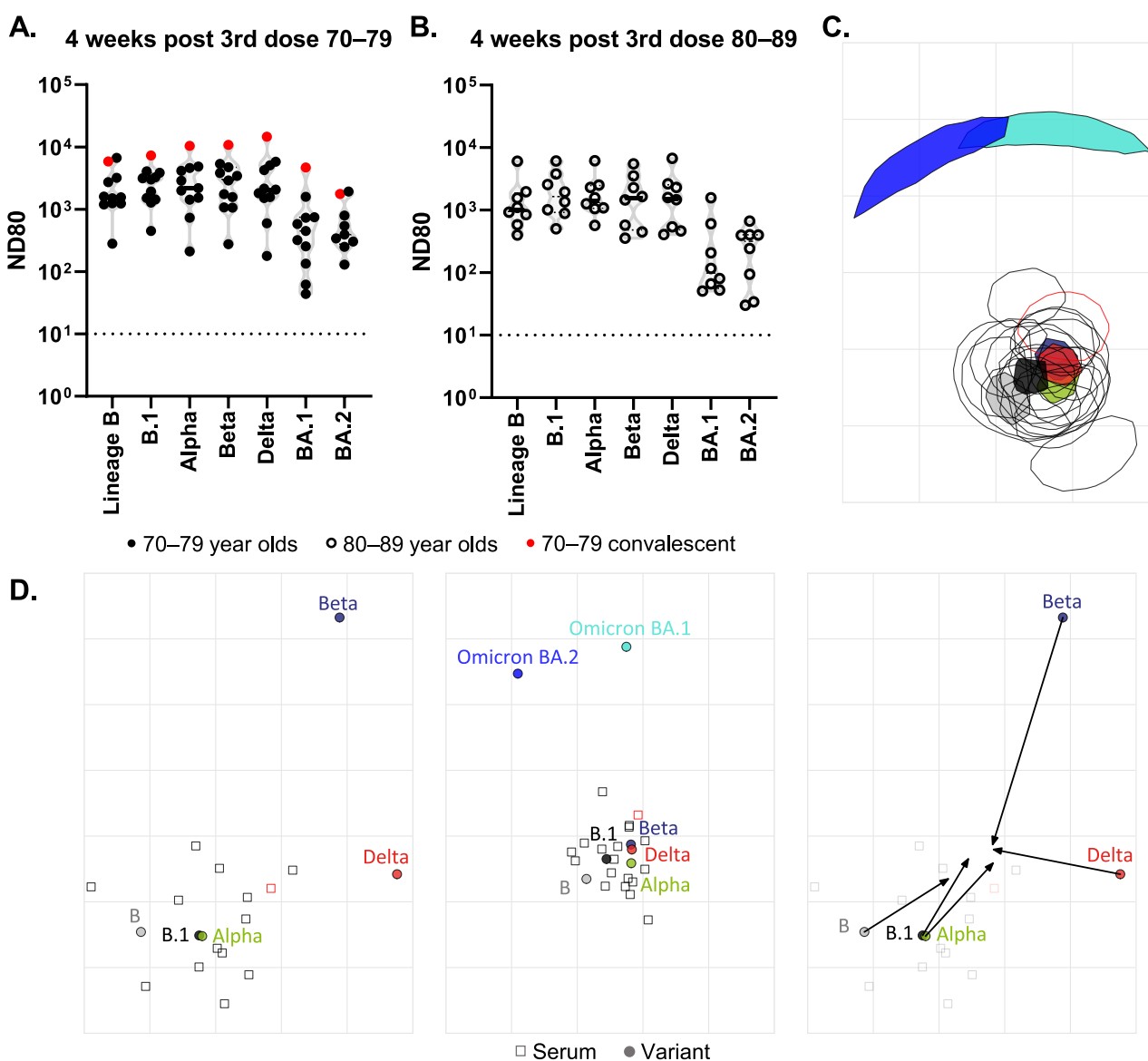

**A.** 4 weeks post 3rd dose 70–79

**B.** 4 weeks post 3rd dose 80–89

**C.**

- • 70–79 year olds
- ○ 80–89 year olds
- • 70–79 convalescent

**D.**

□ Serum  ● Variant

**Extended Data Fig. 4 | Antigenic cartography for 3rd dose samples. a,b**, Neutralisation of pseudotypes bearing the SARS-CoV-2 D614 (Lineage B), B.1 (D614G), Alpha, Beta, Delta or Omicron (BA.1 and BA.2) Spike were compared in the two age-stratified cohorts, 70–79 (solid circles, $n=11$ biologically independent samples) (**A**) and 80–89 (open circles, $n=8$ biologically independent samples) (**b**) from sera collect at 4-weeks post 3rd dose. Symbols in red represents samples taken from individuals who tested positive for SARS-CoV-2 Nucleoprotein by ELISA, indicative of previous infection. Titres are expressed as serum fold-dilution required to achieve 80% virus neutralisation, with the titre ($ND_{80}$) calculated by *xy* interpolation from the mVNT data series (dilution, *x* versus luciferase activity, relative light units, *y*). The detection limit of the assay is defined as a titre of 30 and is indicated with a dotted line. The upper limit of detection is defined as a titre is 21,870. (**c**) Two-dimensional antigenic map of selected variants (the same map as in Fig. 4f, turned 270°), with coordination confidence areas reflecting the uncertainty in positioning of variants and sera. Variants are represented by solid shapes, sera by open shapes (Nucleoprotein-ELISA-positive in red). Each shape encompasses the area on the map that the point could be located at, without increasing the error of the map by more than 1. The spacing between grid lines represents one antigenic unit, equivalent to a two-fold dilution in $ND_{80}$ titres.

**d**, Comparison of maps for post 2nd dose (left-most map, same as Fig. 3c but pruned to show selected variants only) and post 3rd dose (middle map, same as Fig. 4f but turned 270°) $ND_{80}$ data. Arrows in the right-most map highlight the differences in positions of selected variants between the two maps, as determined by a Procrustes transformation: maps were rotated and translated to minimise the error between them, before overlaying one map onto the other. The spacing between grid lines represents one antigenic unit, equivalent to a two-fold dilution in $ND_{80}$ titres, in all maps.

DBPR
NMICROBIOL-21112852B

# Reporting Summary

## Statistics

For all statistical analyses, confirm that the following items are present in the figure legend, table legend, main text, or Methods section.

| n/a | Confirmed | |
|---|---|---|
| ☐ | ☒ | The exact sample size (*n*) for each experimental group/condition, given as a discrete number and unit of measurement |
| ☐ | ☒ | A statement on whether measurements were taken from distinct samples or whether the same sample was measured repeatedly |
| ☐ | ☒ | The statistical test(s) used AND whether they are one- or two-sided<br>*Only common tests should be described solely by name; describe more complex techniques in the Methods section.* |
| ☒ | ☐ | A description of all covariates tested |
| ☐ | ☒ | A description of any assumptions or corrections, such as tests of normality and adjustment for multiple comparisons |
| ☐ | ☒ | A full description of the statistical parameters including central tendency (e.g. means) or other basic estimates (e.g. regression coefficient) AND variation (e.g. standard deviation) or associated estimates of uncertainty (e.g. confidence intervals) |
| ☐ | ☒ | For null hypothesis testing, the test statistic (e.g. *F*, *t*, *r*) with confidence intervals, effect sizes, degrees of freedom and *P* value noted<br>*Give P values as exact values whenever suitable.* |
| ☒ | ☐ | For Bayesian analysis, information on the choice of priors and Markov chain Monte Carlo settings |
| ☒ | ☐ | For hierarchical and complex designs, identification of the appropriate level for tests and full reporting of outcomes |
| ☐ | ☒ | Estimates of effect sizes (e.g. Cohen's *d*, Pearson's *r*), indicating how they were calculated |

*Our web collection on statistics for biologists contains articles on many of the points above.*

## Software and code

Policy information about availability of computer code

| Data collection | GloMax Discover System v3.2.3; |
|---|---|
| Data analysis | Excel v2111 (Microsoft 365); GraphPad Prism v9 (GraphPad Software); Racmacs v1.1.12 (R package) |

For manuscripts utilizing custom algorithms or software that are central to the research but not yet described in published literature, software must be made available to editors and reviewers. We strongly encourage code deposition in a community repository (e.g. GitHub). See the Nature Portfolio guidelines for submitting code & software for further information.

## Data

Policy information about availability of data

All manuscripts must include a data availability statement. This statement should provide the following information, where applicable:

- Accession codes, unique identifiers, or web links for publicly available datasets
- A description of any restrictions on data availability
- For clinical datasets or third party data, please ensure that the statement adheres to our policy

The datasets generated during and/or analysed during the current study are available from the corresponding author on reasonable request. Unique identifiers cannot be provided for the CONSENSUS trial data for individual participants, due to participant confidentiality, as stated in the PHE/UKHSA study approval documentation sections 4.3.5 and 7.1.

# Field-specific reporting

Please select the one below that is the best fit for your research. If you are not sure, read the appropriate sections before making your selection.

☒ Life sciences        ☐ Behavioural & social sciences        ☐ Ecological, evolutionary & environmental sciences

For a reference copy of the document with all sections, see nature.com/documents/nr-reporting-summary-flat.pdf

# Life sciences study design

All studies must disclose on these points even when the disclosure is negative.

| | |
|---|---|
| Sample size | This sample size was not based on a power calculation as all available samples (n=37) were included in the initial section of the study with VOCs. A smaller representative pool of sera from the same cohort (n=16 total) was selected for wider analysis by ranking the neutralisation responses of the whole cohort based on the Wuhan/Beta IC80 ratio, then selecting evenly ranked samples excluding the samples with the highest and lowest ratio. |
| Data exclusions | Two sera were excluded from cartography mapping as their titres were consistently below the predetermined detection limit. |
| Replication | A D614G pseudotype control was included in every neutralisation experiment as a reference, replicated a total of 5 times; these independent repeats showed good concordance and robust repeatability between experiments. Experiments for other individual variants were not replicated due to the finite availability of samples, but were run in biological triplicates within each experiment. |
| Randomization | We tested every available sera from the control arm of the CONSENSUS study in our initial experiments (n=37) so there was no possibility to choose a random selection. For subsequent experiments using smaller pools of sera, we state in the manuscript the randomization process "Using a smaller pool of sera from the same cohort (3-weeks post 2nd dose; n=16 total; 70-79, n=11; 80-89 n=5), selected by ranking the neutralisation ND80 ratio of the lineage B virus to Beta across the whole cohort and picking evenly ranked serum samples" |
| Blinding | Blinding was not possible or applicable to this study as experiments were performed in an unbiased manner. Researchers were unaware of an expected outcome, so would not be able to influence it. |

# Reporting for specific materials, systems and methods

We require information from authors about some types of materials, experimental systems and methods used in many studies. Here, indicate whether each material, system or method listed is relevant to your study. If you are not sure if a list item applies to your research, read the appropriate section before selecting a response.

### Materials & experimental systems

| n/a | Involved in the study |
|---|---|
| ☐ | ☒ Antibodies |
| ☐ | ☒ Eukaryotic cell lines |
| ☒ | ☐ Palaeontology and archaeology |
| ☒ | ☐ Animals and other organisms |
| ☐ | ☒ Human research participants |
| ☒ | ☐ Clinical data |
| ☒ | ☐ Dual use research of concern |

### Methods

| n/a | Involved in the study |
|---|---|
| ☒ | ☐ ChIP-seq |
| ☒ | ☐ Flow cytometry |
| ☒ | ☐ MRI-based neuroimaging |

## Antibodies

| | |
|---|---|
| Antibodies used | Anti-human IgG horseradish peroxidase antibody conjugate  (AP112P, Sigma-Aldrich, Poole, UK), diluted 1:15,000 |
| Validation | *Describe the validation of each primary antibody for the species and application, noting any validation statements on the manufacturer's website, relevant citations, antibody profiles in online databases, or data provided in the manuscript.* |

## Eukaryotic cell lines

Policy information about cell lines

| | |
|---|---|
| Cell line source(s) | HEK293T (ATCC CRL-11268G-1), Vero-E6-TMPRSS2 (King's College London, Stuart JD Neil) |
| Authentication | Cell lines were not authenticated, although they were purchased from ATCC orginally. The morphology of the cell lines appear as expected. |
| Mycoplasma contamination | Cell lines tested negative for mycoplasma contamination. |

| Commonly misidentified lines<br>(See ICLAC register) | None used. |
| --- | --- |

# Human research participants

Policy information about studies involving human research participants

| Population characteristics | We are only able to provide aggregated data for this cohort due to patient confidentiality. We have provided the aggregated information in supplementary datasheet 1. The media age of this cohort was 78 [75-80 IQR], we had a make-up of 54.1% females, 45.9% males. Information about the dates of COVID-19 vaccine are provided in supplementary datasheet 1. |
| --- | --- |
| Recruitment | The CONSENSUS study aimed to recruit healthy participants through London Primary Care Networks (PCN) in 10-year age bands prioritising those over 60 years old. Sera for this study were from some of the first vaccinees in the UK, where participants had received 2 dose doses of  Pfizer-BioNTech vaccine 3 weeks apart (control arm).<br><br>Participants were recruited on a voluntary basis when attending a vaccination clinic vaccination. We do not expect the recruitment to introduce a bias in the analyses described int the paper. |
| Ethics oversight | The protocol was approved by Public Health England Research Ethics Governance Group (reference NR0253; 18/01/21).<br><br>Participants who were unable to provide informed written consent were excluded from the recruitment process. |

Note that full information on the approval of the study protocol must also be provided in the manuscript.

