## [Peer Review File · Nature Microbiology]

Peer Review Information

Journal: Nature Microbiology

Manuscript Title: Neutralising antibody activity against 21 SARS-CoV-2 variants in older adults vaccinated with BNT162b2

Corresponding author name(s): Dalan Bailey

Reviewer Comments & Decisions:

Decision Letter, initial version:

Dear Dr Bailey

Thank you very much for your enquiry about submitting a manuscript to Nature Microbiology.

I've now had a chance to discuss your work with my colleagues, and although we think that it sounds very interesting, we are still uncertain as to the degree to which the study provides novel insights and therefore whether it sufficiently advances the field to be of broad interest.

Therefore, we would like to invite you to submit the full manuscript to Nature Microbiology so that we can fully examine the data before deciding whether to send the paper out to review.

If this is acceptable to you, you can submit the complete manuscript using the link below:

{redacted}

If you have any questions, please feel free to contact me.

Yours sincerely

{redacted}

Decision Letter, first revision:

Dear Dalan,

Thank you for your patience while your manuscript "Neutralising antibody activity against SARS-CoV-2 variants, including Omicron, in an elderly cohort vaccinated with BNT162b2" was under peer-review at Nature Microbiology and my apologies again for the delay in the process. It has now been seen by 3 referees, whose expertise and comments you will find at the of this email. You will see from their comments below that while they find your work of interest, some important points are raised. We are very interested in the possibility of publishing your study in Nature Microbiology, but would like to consider your response to these concerns in the form of a revised manuscript before we make a final

2decision on publication.

In particular, you will see that reviewer #2 asks to include an individual longitudinal trajectory of the antibody response over time for WT and omicron, which we would like to ask you to include if possible in addition to other data and/or analyses that reviewers suggest to include. In addition, given that this topic is very time sensitive we will need your responses and revision back as fast as possible. Please let me know if this will be feasible for you. The rest referees' reports are clear and the remaining issues should be straightforward to address.

If you have not done so already please begin to revise your manuscript so that it conforms to our Article format instructions at <http://www.nature.com/nmicrobiol/info/final-submission/>

The usual length limit for a Nature Microbiology Article is six display items (figures or tables) and 3,000 words. We have some flexibility, and can allow a revised manuscript at 3,500 words, but please consider this a firm upper limit. There is a trade-off of ~250 words per display item, so if you need more space, you could move a Figure or Table to Supplementary Information.

Some reduction could be achieved by focusing any introductory material and moving it to the start of your opening 'bold' paragraph, whose function is to outline the background to your work, describe in a sentence your new observations, and explain your main conclusions. The discussion should also be limited. Methods should be described in a separate section following the discussion, we do not place a word limit on Methods.

Nature Microbiology titles should give a sense of the main new findings of a manuscript, and should not contain punctuation. Please keep in mind that we strongly discourage active verbs in titles, and that they should ideally fit within 90 characters each (including spaces).

Please include a data availability statement as a separate section after Methods but before references, under the heading "Data Availability". This section should inform readers about the availability of the data used to support the conclusions of your study. This information includes accession codes to public repositories (data banks for protein, DNA or RNA sequences, microarray, proteomics data etc...), references to source data published alongside the paper, unique identifiers such as URLs to data repository entries, or data set DOIs, and any other statement about data availability. At a minimum,

2you should include the following statement: "The data that support the findings of this study are available from the corresponding author upon request", mentioning any restrictions on availability. If DOIs are provided, we also strongly encourage including these in the Reference list (authors, title, publisher (repository name), identifier, year). For more guidance on how to write this section please see:

<http://www.nature.com/authors/policies/data/data-availability-statements-data-citations.pdf>

To improve the accessibility of your paper to readers from other research areas, please pay particular attention to the wording of the paper's opening bold paragraph, which serves both as an introduction and as a brief, non-technical summary in about 150 words. If, however, you require one or two extra sentences to explain your work clearly, please include them even if the paragraph is over-length as a result. The opening paragraph should not contain references. Because scientists from other sub-disciplines will be interested in your results and their implications, it is important to explain essential but specialised terms concisely. We suggest you show your summary paragraph to colleagues in other fields to uncover any problematic concepts.

If your paper is accepted for publication, we will edit your display items electronically so they conform to our house style and will reproduce clearly in print. If necessary, we will re-size figures to fit single or double column width. If your figures contain several parts, the parts should form a neat rectangle when assembled. Choosing the right electronic format at this stage will speed up the processing of your paper and give the best possible results in print. We would like the figures to be supplied as vector files - EPS, PDF, AI or postscript (PS) file formats (not raster or bitmap files), preferably generated with vector-graphics software (Adobe Illustrator for example). Please try to ensure that all figures are non-flattened and fully editable. All images should be at least 300 dpi resolution (when figures are scaled to approximately the size that they are to be printed at) and in RGB colour format. Please do not submit Jpeg or flattened TIFF files. Please see also 'Guidelines for Electronic Submission of Figures' at the end of this letter for further detail.

Figure legends must provide a brief description of the figure and the symbols used, within 350 words, including definitions of any error bars employed in the figures.

When submitting the revised version of your manuscript, please pay close attention to our [href="https://www.nature.com/nature-research/editorial-policies/image-integrity">Digital Image Integrity Guidelines](https://www.nature.com/nature-research/editorial-policies/image-integrity) and to the following points below:

3Please include a statement before the acknowledgements naming the author to whom correspondence and requests for materials should be addressed.

Finally, we require authors to include a statement of their individual contributions to the paper -- such as experimental work, project planning, data analysis, etc. -- immediately after the acknowledgements. The statement should be short, and refer to authors by their initials. For details please see the Authorship section of our joint Editorial policies at http://www.nature.com/authors/editorial_policies/authorship.html

- * include a point-by-point response to any editorial suggestions and to our referees. Please include your response to the editorial suggestions in your cover letter, and please upload your response to the referees as a separate document.

- * ensure it complies with our format requirements for Letters as set out in our guide to authors at www.nature.com/nmicrobiol/info/gta/

- * state in a cover note the length of the text, methods and legends; the number of references; number and estimated final size of figures and tables

- * resubmit electronically if possible using the link below to access your home page:

{redacted}

- *This url links to your confidential homepage and associated information about manuscripts you may have submitted or be reviewing for us. If you wish to forward this e-mail to co-authors, please delete this link to your homepage first.

Please ensure that all correspondence is marked with your Nature Microbiology reference number in the subject line.

Nature Microbiology is committed to improving transparency in authorship. As part of our efforts in this direction, we are now requesting that all authors identified as 'corresponding author' on published papers create and link their Open Researcher and Contributor Identifier (ORCID) with their account on the Manuscript Tracking System (MTS), prior to acceptance. This applies to primary research papers only. ORCID helps the scientific community achieve unambiguous attribution of all scholarly contributions. You can create and link your ORCID from the home page of the MTS by clicking on 'Modify my Springer Nature account'. For more information please visit www.springernature.com/orcid.

We hope to receive your revised paper within three weeks. If you cannot send it within this time, please let us know.

Yours sincerely,

{redacted}

Reviewer Expertise:

Referee #1: vaccine response, clinician
Referee #2: geriatrics, epidemiology
Referee #3: response to vaccines and infections

Reviewers Comments:

Reviewer #1 (Remarks to the Author):

Neutralizing antibody activity against SARS-CoV-2 variants, including Omicron, in an elderly cohort vaccinated with BNT162b2 by J. Newman et al.

Main points:

- Study conducted in North London, n=37, appears powered to make comparisons between age cohorts.
- The pseudotype-based microneutralization assay is robustly repeatable with minimal intra-assay variation (data in supplement).
 - o When vaccine virus and target virus are well-matched, a neutralization assay correlates well with ELISA results, but as target virus diverges, this correlation disappears.
- Neutralization titers in elderly cohorts decrease between 3 and 20 weeks post-second vaccination.
 - o The older cohort had lower maximal neutralization titers than the younger of two elderly cohorts.
- D614G and Alpha variants had minimal drop in sensitivity to neutralization, with more decrease seen for Delta and strong decrease for Beta. Beta is the most antigenically distinct variant virus, primarily due to spike mutations at position 484. Authors suggest that looking for viruses bearing this mutation can help predict risk of future VOCs.
- Virtually no anti-Omicron variant virus neutralizing activity is seen in elderly subject's serum until after a third dose of the BNT162b2 vaccine
- Novel components:
 - o Uniquely aged population @ 70-89 (rather than just >50)
 - o Antigenic cartography shows that the VOCs and VOIs are antigenically variable
 - o Testing against 17 variants
 - o Antibody responses after a third vaccine dose
- Neutralizing antibody titers in the elderly 3-12 weeks post second vaccination have been previously

5reported for beta, alpha and gamma:

o Collier, D.A., Ferreira, I.A.T.M., Kotagiri, P. et al. Age-related immune response heterogeneity to SARS-CoV-2 vaccine BNT162b2. *Nature* 596, 417–422 (2021). <https://doi.org/10.1038/s41586-021-03739-1>

• ELISA titers after doses 1 and two from the same cohort have already been reported:

o Subbarao Sathyavani, Warren Lenesh A, Hoschler Katja, Perry Keith R, Shute Justin, Whitaker Heather, O'Brien Michelle, Baawuah Frances, Moss Paul, Parry Helen, Ladhani Shamez N, Ramsay Mary E, Brown Kevin E, Amirthalingam Gayatri. Robust antibody responses in 70–80-year-olds 3 weeks after the first or second doses of Pfizer/BioNTech COVID-19 vaccine, United Kingdom, January to February 2021. *Euro Surveill.* 2021;26(12):pii=2100329. <https://doi.org/10.2807/1560-7917.ES.2021.26.12.2100329>

Other Concerns:

• Why does the paper shift from using D614-specific responses as a baseline of comparison to using D614G-specific responses as a baseline of comparison? This switch does not appear to be explained. It appears that D614G-specific neutralization titers are higher than D614, although D614 was the vaccination protein. Can the authors speculate on why titers are higher to the mutant? Are the higher titers to the mutant the reason D614G was later used as a point of comparison? Does this choice conflate the drops in neutralization titers reported?

• Smaller representative panel of serum was selected by ranking the neutralizing antibody responses of the whole cohort – and then what?

• The methods lacks a statistical analysis section.

• P values of Spearman correlations should be added to correlation graphs

• Figure 2A is too small to be informative.

• Figure 2B and 2C should be shown with the same y axis, and figure 2E and 2F should be shown with the same y axis.

• What is the point the map in figure 3A? Please explain in the caption what the various sized dots are meant to indicate – or remove the map from the figure altogether. It is not clear what value the map adds to the manuscript.

Minor comments:

• Lines 83-92 should be updated with more recent information: this was written before Omicron became widespread.

• Include (Theta) in P.3 designation of figure 3C

• Line 181: Correct Figure 2B-C to Figure 2B

Reviewer #2 (Remarks to the Author):

The paper by Newman et al investigates antibody activity against SARS-CoV-2 variants in an elderly cohort with repeated measurements from the same individuals over time. They also add data on the latest omicron variant. They show that neutralizing antibodies wane over time, and although this is also suggested by many other groups, this paper adds knowledge on many different variants and with regards to the elderly population. Moreover, repeated measurements make it possible to follow the same individuals over time and to do matched analyses. The paper is nicely written, some improvements suggested below.

1. A statistical methods section is missing. Please describe all statistical analyses here. Would be good to include mean and SD of all groups as well.
2. Are mean levels overall lower in the oldest group compared to the younger?
3. Any other information available on the samples? Sex, multi-morbidity, medication etc? If so, please consider it in analyses and provide information in a table.
4. Is it possible to do individual longitudinal trajectories of the antibody response over time (WT and omicron)? From 3 weeks post 2nd dose, to 20 weeks, to 4 weeks post 3rd dose? Please add if possible.
5. Another reference that may be important to cite: Neutralization and Stability of SARS-CoV-2 Omicron Variant (nih.gov), PMID: 34981053
6. Is the booster same as the 1st and 2nd dose vaccine (Pfizer/BioNTech)?

Reviewer #3 (Remarks to the Author):

Newman and colleagues investigated the neutralizing Ab(nAb) responses elicited in a cohort (N=37) of elderly (70–90-year-old) subjects that received 2 or 3 doses of the Pfizer vaccine. They examined whether this vaccine elicited nAb responses not only against the 'original' SARS-CoV-2 strain (homologous to the vaccine) but also against several SARS-CoV-2 variants and they determined the titers and durability of these responses. Correlations between binding Abs to the S and RBD derived from different variants and nAbs were performed and efforts were made to identify the escape mutations in S. They report that the third immunization (boost) with the same vaccine is necessary to elicit nAb responses against variants of concern, in particular Omicron. The topic of this work is obviously timely, and although others have reported on the lower nAb titers elicited by mRNA vaccines against VOCs, I believe this is the first study that specifically examined how elderly subjects respond to mRNA vaccination. Overall, the experiments are well conducted, and the statistical analyses are appropriate.

They report that the nAb responses elicited by 2 doses of this vaccine wane over time and that they are not effective against most VOC, especially Omicron (as other studies have recently shown), but that a third dose of the same vaccine results in a boosting of the nAb responses overall, including those against Omicron. The study therefore indicates that elderly subjects will benefit from a regiment of three immunizations with the Pfizer vaccine.

It would be interesting to examine whether the nAb titers after the third immunization decrease with the same kinetics as those elicited after the 2nd immunization, if appropriate samples are available.

7Against the Wuhan strain (homologous to the vaccine), they report that (a) a significant drop of nAb titers occurs between 3 and 20 weeks post 2nd dose and (b) that subjects aged 80-90 have in general lower titers at 3 and 20 weeks post 2nd dose than subjects aged 70-79. Could the authors comment on whether the differences in nAb titers were statistically different between the two groups at either timepoint?

They report that the nAbs elicited at 3 weeks post the 2nd dose in both the 70-79 and 80-89 subgroups, are much lower against the delta and beta VOCs than against the D614G variants. Were similar significant differences also observed between the WT D614 strain and these VOCs?

In lines 145-147, the authors argue that there was 'evidence of affinity maturation in some individuals' of the nAbs between 3 and 20 weeks post 2nd vaccination; especially against the Beta VOC. However, changes in anti-Beta nAb titers between 3 and 20 weeks post 2nd immunization (SupFig 2) could be due to a delayed emergence of a particular subset of nAbs, rather than a true 'affinity' maturation of nAbs. The most direct way to address this important point would be to isolate S-specific B cells at 3 and 20 weeks post immunization and sequence their VH/VL genes.

A very strong part of this study is the analysis of nAb responses not only against the 5 VOCs, but against 15 other variants (including variants under investigation and variants under monitoring). Importantly the authors were able to identify specific mutations in S that appear to be responsible for escape from nAbs elicited by the Pfizer vaccine. They propose that most likely a combination of mutations is responsible for the escape phenotype of certain VOCs. They did not however prove this experimentally (by targeted mutagenesis).

The analysis of the anti-Omicron nAbs is also a strong part of this study. The results indicate that 2 immunizations with this vaccine do not generally elicit detectable anti-Omicron nAbs (something that others have already reported), but the third (boost) immunization results in anti-Omicron nAbs in all subjects examined. The authors mention that at 3 weeks post the 2nd immunization two out of twelve 70-79-year-old had anti-Omicron nAbs. However, on Fig 4B this does not appear to be the case. The ND80 values appear to be at the background level. Could the authors address this point?

Author Rebuttal, first revision:

Reviewer #1:

Neutralizing antibody titers in the elderly 3-12 weeks post second vaccination have been previously reported for beta, alpha and gamma: Collier, D.A., Ferreira, I.A.T.M., Kotagiri, P. et al. Age-related immune response heterogeneity to SARS-CoV-2 vaccine BNT162b2. Nature 596, 417-422 (2021). ELISA titers after doses 1 and two from the same cohort have already been reported: Subbarao Sathyavani, Warren Lenessa A, Hoschler Katja, Perry Keith R, Shute Justin, Whitaker Heather, O'Brien Michelle, Baawuah Frances, Moss Paul, Parry Helen, Ladhani Shamez N, Ramsay Mary E, Brown Kevin E, Amirthalingam Gayatri. Robust antibody responses in 70-80-year-olds 3 weeks after the first or second doses of Pfizer/BioNTech COVID-19 vaccine, United Kingdom, January to February 2021. Euro Surveill. 2021;26(12):pii=2100329.

We acknowledge the prior publication of related papers on neutralising antibody responses in the elderly; however, we hope the addition of our data and its expansion to novel variants, including BA.1/BA.2, as well as post-3rd dose titres, provides sufficient novelty. Whilst some of the ELISA data from our paper has been published elsewhere, we acknowledge this in the manuscript, and have only used this data for correlations with novel VNT data, data which was previously unavailable. We have also built on this ELISA data (Roche commercial assay) by including VOC-specific in-house RBD-ELISA data in the manuscript.

Why does the paper shift from using D614-specific responses as a baseline of comparison to using D614G-specific responses as a baseline of comparison? This switch does not appear to be explained. It appears that D614G-specific neutralization titers are higher than D614, although D614 was the vaccination protein. Can the authors speculate on why titers are higher to the mutant? Are the higher titers to the mutant the reason D614G was later used as a point of comparison? Does this choice conflate the drops in neutralization titers reported?

The variants are compared to D614G, rather than D614 (Wuhan) as from Figure 2 onwards we were trying to address the epidemiological significance of each variant, relative to the first wave virus which was D614G or B.1. Within the UKRI-funded Genotype-to-phenotype consortium (with which we are working), and more widely, this is considered a more appropriate comparator as most hospitalisation and death data etc. is based on this “variant”, with the same being true of the major VOCs that followed, Alpha, Delta etc (since D614 hardly emerged from China). The reason we specifically examined D614/Wuhan specific responses in Figure 1 was to measure the overall response (and waning) to homologously matched immunogens/antigens, more as a fair measure of the immune response. As to why the titers are higher, other labs have seen similar findings – the general consensus being that the receptor binding domains (RBDs) of D614G are more likely to be an up/open conformation than in D614/Wuhan. The net result is increased infectivity, but also increased sensitivity to neutralising antibodies, the majority of which target the RBD. To simplify this, within the manuscript we have removed the Wuhan data from Figure 2B and 2C, which was repeated from Figure 1. In addition, to highlight the epidemiological significance of the comparisons throughout the paper we now refer to D614G as B.1, which is its correct Pango lineage (since it does not have a Greek letter assigned).

Smaller representative panel of serum was selected by ranking the neutralizing antibody responses of the whole cohort – and then what?

We apologise for the confusion – this panel selection was not properly described. This has now been remedied in the text, as follows, “Using a smaller pool of sera from the same cohort (3-weeks post 2nd dose; n=16 total; 70-79, n=11; 80-89 n=5), selected by ranking the neutralisation ND80 ratio of Wuhan

to Beta across the whole cohort and picking evenly ranked serum samples, we widened our analysis to fifteen other SARS-CoV-2 variants, including other VUIs and VUMs". The purpose for this smaller selection and ranking mechanism was two-fold; a) to preserve sera stocks and b) to rank and identify sera which responded differently to antigenically variable variants, using Beta and Wuhan as the comparators.

The methods lacks a statistical analysis section.

A statistical methods section has now been added, outlining the relevant statistical tests used throughout the manuscript (median (LQ-UQ), Wilcoxon Matched-Pairs signed Rank test, Mann-Whitney test, Spearman's rank correlation, Friedman test and Dunn's test for multiple comparisons). These tests were performed using GraphPad Prism 9 and Microsoft Excel.

P values of Spearman correlations should be added to correlation graphs

These have been added to all correlation graphs as requested.

Figure 2A is too small to be informative.

We have increased the size of Figure 2A to improve readability.

Figure 2B and 2C should be shown with the same y axis, and figure 2E and 2F should be shown with the same y axis.

Apologies, this was an oversight caused by re-formatting prior to submission. These graphs have now been plotted with the same matched axes.

What is the point the map in figure 3A? Please explain in the caption what the various sized dots are meant to indicate – or remove the map from the figure altogether. It is not clear what value the map adds to the manuscript.

The map has been moved to the supplemental data. It's purpose is simply to highlight where individual variants were first detected, to illustrate the global nature of the problem and to highlight the convergent evolution of similar changes, e.g. Spike E484K, in geographically isolated locations.

Minor comments:

Lines 83-92 should be updated with more recent information: this was written before Omicron became widespread.

The introduction has now been updated, to acknowledge more recent publications on Omicron, the ongoing emergence of BA.2 and to meet the word limit of Nature Microbiology.

Include (Theta) in P.3 designation of figure 3C.

This has been added as requested.

Line 181: Correct Figure 2B-C to Figure 2B

This is the correct Figure reference. We are making comparisons between the reduction in neutralisation with various VUIs to the Beta VOC data in Figure 2. In Figure 3 we used a smaller pool of sera where the age groups were mixed. However, these samples can still be paired to those used in Figure 2. The value quoted (9/16; 56.3%) represents the Beta values for people from age groups 70-79 and 80-89 hence the referral to Figure 2B and 2C.

Reviewer #2:

A statistical methods section is missing. Please describe all statistical analyses here. Would be good to include mean and SD of all groups as well.

A statistical methods section has now been added, outlining the relevant statistical tests used throughout the manuscript (median (LQ-UQ), Wilcoxon Matched-Pairs signed Rank test, Mann-Whitney test, Spearman's rank correlation, Friedman test and Dunn's test for multiple comparisons). These tests were performed using GraphPad Prism 9 and Microsoft Excel. For the average and variation comparison between groups, we have used median and the lower/upper quartile, given the large outliers in the datasets in non-responders (i.e., ND80 ≤ 10) or in Nucleoprotein-ELISA-positive individuals (i.e., ND80 ≥ 2560 or ≥ 7290), which could skew the mean. However, the mean and SD of all groups has been summarised in the Supplemental Datasets together with other statistical analysis of the data, e.g. median fold changes, Mann-Whitney age group comparisons etc.

Are mean levels overall lower in the oldest group compared to the younger?

11We have now added comparisons between the two aged groups (70-79 and 80-89); as median fold-change relative to the 70-79 age group. At 3-weeks post second dose, we see that the Ab titres are generally lower in the 80-89 age group against the WT/D614, B.1/D614G, B.1.1.7/Alpha and B.1.617.2/Delta variants. However, when the titres are low in both age groups e.g., at 3-week post 2nd dose against B.1.351/Beta and Omicron (BA.1/BA.2) or at 20-weeks post second dose, this difference is less marked. At 4-weeks post 3rd dose, there are between 1.2 – 1.9-fold lower nAb titres detected in the 80-89 age group compared to the 70-79 age group (WT: 1.5-fold, B.1: 1.8-fold, Alpha: 1.5-fold, Beta:1.9-fold, Delta: 1.4-fold, BA.2: 1.2-fold), with the only large difference seen against BA.1 (4.5-fold decrease). These comparisons have now been included in the text in the results section and summarised in the Supplemental Datasets (as mentioned above).

Any other information available on the samples? Sex, multi-morbidity, medication etc? If so, please consider it in analyses and provide information in a table.

We have added the aggregated (>5) volunteer-disclosed information on sex and age as well as times between vaccination to the anonymised source dataset (Supplemental Datasets) that will accompany this submission. Unfortunately, we are unable to report on individual co-morbidities.

Is it possible to do individual longitudinal trajectories of the antibody response over time (WT and omicron)? From 3 weeks post 2nd dose, to 20 weeks, to 4 weeks post 3rd dose? Please add if possible.

We have now added longitudinal trajectories for B.1, BA.1 and now BA.2 ND80s in Figure 4, updating the dataset to reflect the current epidemiological situation and to include relevant RBD-ELISA data and cartography for Omicron and other VOCs post 3rd dose.

Another reference that may be important to cite: Neutralization and Stability of SARS-CoV-2 Omicron Variant (nih.gov), PMID: 34981053

This reference has been cited.

Is the booster same as the 1st and 2nd dose vaccine (Pfizer/BioNTech)?

Yes, all three doses received by this cohort are Pfizer/BioNTech BNT162b2. This information is included in the “participant and ethical statement” section of the methods: “Sera samples used in this study were taken at 3 and 20 weeks after 2 doses of Pfizer/BioNTech BNT162b2 (Mainz, Germany) – a lipid nanoparticle-formulated, nucleoside-modified RNA vaccine encoding prefusion stabilized SARS-CoV-2 spike COVID-19 vaccine given at a 3-week interval, as well as 4 weeks post 3rd dose” and has now been clarified further in the introduction and results sections.

12Reviewer #3:

It would be interesting to examine whether the nAb titers after the third immunization decrease with the same kinetics as those elicited after the 2nd immunization, if appropriate samples are available.

Unfortunately, we do not yet have access to these samples; however, we plan a follow-up study to address the reviewer's comment in the Summer of 2022.

Against the Wuhan strain (homologous to the vaccine), they report that (a) a significant drop of nAb titers occurs between 3 and 20 weeks post 2nd dose and (b) that subjects aged 80-90 have in general lower titers at 3 and 20 weeks post 2nd dose than subjects aged 70-79. Could the authors comment on whether the differences in nAb titers were statistically different between the two groups at either timepoint?

In most cases the median ND80s in the 80-89 age group are lower than in the 70-79 age group. We have now made additional comparisons of these groups in the results section. In most cases however the differences were not significantly different following a Mann-Whitney test for statistical significance (likely because these samples are not paired). The results of these comparisons and tests have been made available in the Supplemental Datasets.

They report that the nAbs elicited at 3 weeks post the 2nd dose in both the 70-79 and 80-89 subgroups, are much lower against the delta and beta VOCs than against the D614G variants. Were similar significant differences also observed between the WT D614 strain and these VOCs?

As discussed above, in response to reviewer 1's related question on D614G vs D614 comparisons, our main intention was to first examine nAb responses to the matched immunogen/antigen as well as waning (Figure 1) before addressing the epidemiologically wave-specific differences in neutralisation, e.g. D614G vs. Alpha or Beta (Figure 2). This is because most hospitalisation data etc. is available for on these viruses (e.g. D614G/B.1) and not Wuhan/D614. Accordingly, we have removed the repeated Wuhan D614 dataset from Figure 2 and renamed D614G as B.1 to highlight these comparisons. However, to address the reviewer's question we compared the Delta and Beta VOCs at 3 weeks post 2nd dose to the paired Wuhan/D614 titres and they too were significantly different (for both age groups by Wilcoxon matched-pairs signed rank test).

In lines 145-147, the authors argue that there was 'evidence of affinity maturation in some individuals' of the nAbs between 3 and 20 weeks post 2nd vaccination; especially against the Beta

VOC. However, changes in anti-Beat nAb titers between 3 and 20 weeks post 2nd immunization (SupFig 2) could be due to a delayed emergence of a particular subset of nAbs, rather than a true 'affinity' maturation of nAbs. The most direct way to address this important point would be to isolate S-specific B cells at 3 and 20 weeks post immunization and sequence their VH/VL genes.

We agree with the reviewer that elucidating to affinity maturation of nAbs may be a premature assumption due to the lack of specific experimentation, e.g. isolating S-specific B cells. Furthermore, we found differences in the literature on affinity maturation following COVID-19 infection/vaccination. For example, work by Muecksh et al., (doi: [10.1016/j.immuni.2021.07.008](https://doi.org/10.1016/j.immuni.2021.07.008)) provide some evidence that affinity maturation is important, but the failure of SARS-CoV-2 infection to induce B cell stimulation and expansion to generate high nAb titres is also a possibility. Conversely, a preprint by He et al., (doi: [10.1101/2021.09.08.459480](https://doi.org/10.1101/2021.09.08.459480)) suggests that antibody antigen binding features may be germline encoded and the requirement of affinity maturation is limited. Due to the inconclusive evidence from published data and lack of experimentation on our part, we have omitted the sentence referring to affinity maturation from our manuscript.

A very strong part of this study is the analysis of nAb responses not only against the 5 VOCs, but against 15 other variants (including variants under investigation and variants under monitoring). Importantly the authors were able to identify specific mutations in S that appear to be responsible for escape from nAbs elicited by the Pfizer vaccine. They propose that most likely a combination of mutations is responsible for the escape phenotype of certain VOCs. They did not however prove this experimentally (by targeted mutagenesis).

Whilst it is true that we did not perform targeted mutagenesis followed by pseudotyping we did perform targeted mutagenesis followed by RBD-ELISA. This highlighted the important contributory roles of 417 and 484 changes to the reduction in binding seen with a Beta RBD. The role of the SARS-CoV-2 RBD as the major target for neutralising antibodies is well established. Addition of neutralisation data for an E484K pseudotype would perhaps add another level of confirmation to this, but the effect of this mutation on neutralisation is well established in the field, including by us in related research <https://www.biorxiv.org/content/10.1101/2021.08.17.456606v1> - currently in press at {redacted}.

The analysis of the anti-Omicron nAbs is also a strong part of this study. The results indicate that 2 immunizations with this vaccine do not generally elicit detectable anti-Omicron nAbs (something that others have already reported), but the third (boost) immunization results in anti-Omicron nAbs in all subjects examined. The authors mention that at 3 weeks post the 2nd immunization two of twelve 70–79-year-old had anti-Omicron nAbs. However, on Fig 4B this does not appear to be the case. The ND80 values appear to be at the background level. Could the authors address this point?

14We have re-drawn these graphs to show the longitudinal response to B.1, BA.1 and BA.2 (new Figure 4). However, this statement is technically correct, although admittedly a little confusing. The previously infected individual (in red) has a high titre against all variants, with one additional person having a titre of 10.07, which is just over our detection limit (10 ND80).

We hope that these modifications to the manuscript are acceptable and that this work is now suitable for publication in Nature Microbiology. Lastly, we would like to thank the reviewers again for their insightful comments and suggestions regarding our submission.

Decision Letter, second revision:

Dear Dalan,

Thank you for submitting your revised manuscript "Neutralising antibody activity against SARS-CoV-2 variants, including Omicron, in an elderly cohort vaccinated with BNT162b2" (NMICROBIOL-21112852B). It has now been seen by the original referees and their comments are below. The reviewers find that the paper has improved in revision, and therefore we'll be happy in principle to publish it in Nature Microbiology, pending minor revisions to comply with our editorial and formatting guidelines. Please note as well, that reviewer #1 had no further comments or suggestions.

Thank you again for your interest in Nature Microbiology Please do not hesitate to contact me if you have any questions.

Sincerely,

{redacted}

Reviewer #2 (Remarks to the Author):

The authors have answered the reviewer questions nicely and edited the manuscript correspondingly. I have no further comments.

Reviewer #3 (Remarks to the Author):

The authors addressed my comments/concerns appropriately. I do not have additional comments.

Decision Letter, final checks:

Dear Dalan,

Thank you for your patience as we've prepared the guidelines for final submission of your Nature Microbiology manuscript, "Neutralising antibody activity against SARS-CoV-2 variants, including Omicron, in an elderly cohort vaccinated with BNT162b2" (NMICROBIOL-21112852B). Please carefully follow the step-by-step instructions provided in the attached file, and add a response in each row of the table to indicate the changes that you have made. Please also check and comment on any additional marked-up edits we have proposed within the text. Ensuring that each point is addressed will help to ensure that your revised manuscript can be swiftly handed over to our production team.

In recognition of the time and expertise our reviewers provide to Nature Microbiology's editorial process, we would like to formally acknowledge their contribution to the external peer review of your manuscript entitled "Neutralising antibody activity against SARS-CoV-2 variants, including Omicron, in an elderly cohort vaccinated with BNT162b2". For those reviewers who give their assent, we will be publishing their names alongside the published article.

Nature Microbiology offers a Transparent Peer Review option for new original research manuscripts submitted after December 1st, 2019. As part of this initiative, we encourage our authors to support increased transparency into the peer review process by agreeing to have the reviewer comments, author rebuttal letters, and editorial decision letters published as a Supplementary item. When you

16submit your final files please clearly state in your cover letter whether or not you would like to participate in this initiative. Please note that failure to state your preference will result in delays in accepting your manuscript for publication.

Cover suggestions

As you prepare your final files we encourage you to consider whether you have any images or illustrations that may be appropriate for use on the cover of Nature Microbiology.

Nature Microbiology has now transitioned to a unified Rights Collection system which will allow our Author Services team to quickly and easily collect the rights and permissions required to publish your work. Approximately 10 days after your paper is formally accepted, you will receive an email in providing you with a link to complete the grant of rights. If your paper is eligible for Open Access, our Author Services team will also be in touch regarding any additional information that may be required to arrange payment for your article.

Please note that *Nature Microbiology* is a Transformative Journal (TJ). Authors may publish their research with us through the traditional subscription access route or make their paper immediately open access through payment of an article-processing charge (APC). Authors will not be required to make a final decision about access to their article until it has been accepted. [Find out more about Transformative Journals](https://www.springernature.com/gp/open-research/transformative-journals)

Authors may need to take specific actions to achieve [compliance with funder and institutional open access mandates](https://www.springernature.com/gp/open-research/funding/policy-compliance-faqs). If your research is supported by a funder that requires immediate open access (e.g. according to [Plan S principles](https://www.springernature.com/gp/open-research/plan-s-compliance))

17then you should select the gold OA route, and we will direct you to the compliant route where possible. For authors selecting the subscription publication route, the journal's standard licensing terms will need to be accepted, including [self-archiving policies](https://www.nature.com/nature-portfolio/editorial-policies/self-archiving-and-license-to-publish). Those licensing terms will supersede any other terms that the author or any third party may assert apply to any version of the manuscript.

Please use the following link for uploading these materials:
{redacted}

Best regards,

{redacted}

Reviewer #1:
None

Reviewer #2:
Remarks to the Author:
The authors have answered the reviewer questions nicely and edited the manuscript correspondingly. I have no further comments.

Reviewer #3:
Remarks to the Author:
The authors addressed my comments/concerns appropriately. I do not have additional comments.

Final Decision Letter:

Dear Dalan,

Thank you very much for your quick responses to our final requests! I am pleased to accept your Article "Neutralising antibody activity against 21 SARS-CoV-2 variants in older adults vaccinated with BNT162b2" for publication in Nature Microbiology. Thank you for having chosen to submit your work

18to us and many congratulations.

Due to the importance of these deadlines, we ask you to please let us know now whether you will be difficult to contact over the next month. If this is the case, we ask you to provide us with the contact information (email, phone and fax) of someone who will be able to check the proofs on your behalf, and who will be available to address any last-minute problems.

Acceptance of your manuscript is conditional on all authors' agreement with our publication policies (see <https://www.nature.com/nmicrobiol/editorial-policies>). In particular your manuscript must not be published elsewhere and there must be no announcement of the work to any media outlet until the publication date (the day on which it is uploaded onto our website).

Please note that *Nature Microbiology* is a Transformative Journal (TJ). Authors may publish their research with us through the traditional subscription access route or make their paper immediately open access through payment of an article-processing charge (APC). Authors will not be required to make a final decision about access to their article until it has been accepted. [Find out more about Transformative Journals](https://www.springernature.com/gp/open-research/transformative-journals)

Authors may need to take specific actions to achieve [compliance with funder and institutional open access mandates](https://www.springernature.com/gp/open-research/funding/policy-compliance-faqs). If your research is supported by a funder that requires immediate open access (e.g. according to [Plan S principles](https://www.springernature.com/gp/open-research/plan-s-compliance)) then you should select the gold OA route, and we will direct you to the compliant route where possible. For authors selecting the subscription publication route, the journal's standard licensing terms will need to be accepted, including [self-archiving policies](https://www.nature.com/nature-portfolio/editorial-policies/self-archiving-and-license-to-publish). Those licensing terms will supersede any other terms that the author or any third party may assert apply to any version of the manuscript.
